



**Recent acceleration of Denman Glacier (1972-2017), East Antarctica, driven by grounding line retreat and changes in ice tongue configuration.**

Bertie W.J. Miles[1]*, Jim R. Jordan[2], Chris R. Stokes[1], Stewart S.R. Jamieson[1], G. Hilmar Gudmundsson[2], Adrian Jenkins[2]

[1]Department of Geography, Durham University, Durham, DH1 3LE, UK

[2]Department of Geography and Environmental Sciences, Northumbria University, Newcastle upon Tyne, NE1 8ST, UK

*Correspondence to: a.w.j.miles@durham.ac.uk

**Abstract:** Denman Glacier is one of the largest in East Antarctica, with a catchment that contains an ice volume equivalent to 1.5 m of global sea-level and which sits in the Aurora Subglacial Basin (ASB). Geological evidence of this basin's sensitivity to past warm periods, combined with recent observations showing that Denman's ice speed is accelerating, and its grounding line is retreating along a retrograde slope, have raised the prospect that it could contribute to near-future sea-level rise. In this study, we produce the first long-term (~ 50 years) record of past glacier behaviour (ice flow speed, ice tongue structure, and calving) and combine these observations with numerical modelling to explore the likely drivers of its recent change. We find a spatially widespread acceleration of the Denman system since the 1970s across both its grounded (17 ±4% acceleration; 1972-2017) and floating portions (36 ±5% acceleration; 1972-2017). Our numerical modelling experiments show that a combination of grounding line retreat, ice tongue thinning and the unpinning of Denman's ice tongue from a pinning point following its last major calving event are required to simulate an acceleration comparable with observations. Given its bed topography and the geological evidence that Denman Glacier has retreated substantially in the past, its recent grounding line retreat and ice flow acceleration suggest that it could be poised to make a significant contribution to sea level over the coming century.

## 1. Introduction

Over the past two decades, outlet glaciers along the coastline of Wilkes Land, East Antarctica, have been thinning (Pritchard et al., 2009; Flament and Remy, 2012; Helm et al., 2014;



Schröder et al., 2018), losing mass (King et al., 2012; Gardner et al., 2018; Shen et al., 2018;
Rignot et al., 2019) and retreating (Miles et al., 2013; Miles et al., 2016). This has raised
concerns about the future stability of some major outlet glaciers that primarily drain the Aurora
Subglacial Basin (ASB), particularly Totten, Denman, Moscow University and Vanderford
Glaciers (Fig. 1). This is because their present day grounding lines are close to deep retrograde
slopes (Morlighem et al., 2020), meaning there is clear potential for marine ice sheet instability
and future rapid mass loss (Weertman, 1974; Schoof, 2007), unless ice shelves provide a
sufficient buttressing effect (Gudmundsson, 2013). Geological evidence suggests that there
may have been substantial retreat of the ice margin in the ASB during the warm interglacials
of the Pliocene (Williams et al., 2010; Young et al., 2011; Aitken et al., 2016; Scherer et al.,
2016), which potentially resulted in global mean sea level contributions of up to 2 m from the
ASB (Aitken et al., 2016). This is important because these warm periods of the Pliocene may
represent our best analogue for climate by the middle of this century under unmitigated
emission trajectories (Burke et al., 2018). Indeed, numerical models now predict future sea
level contributions from the outlet glaciers which drain the ASB over the coming decades to
centuries (Golledge et al., 2015; Ritz et al., 2015; DeConto and Pollard, 2016), but large
uncertainties exist over the magnitude and rates of any future sea level contributions.
At present, most studies in Wilkes Land have focused on Totten Glacier which is losing mass
(Li et al., 2016; Mohajerani et al., 2019) in association with grounding line retreat (Li et al.,
2015). This has been attributed to wind-forced warm Modified Circumpolar Deep Water
accessing the cavity below Totten Ice Shelf (Greenbaum et al., 2015; Rintoul et al., 2016;
Greene et al., 2017). However, given our most recent understanding of bedrock topography in
Wilkes Land, Denman Glacier provides the most direct pathway to the deep interior of the ASB
(Gasson et al., 2015; Brancato et al., 2020; Morlighem et al., 2020). Moreover, a recent mass
balance estimate (Rignot et al., 2019) has shown that, between 1979 and 2017, Denman
Glacier's catchment may have lost an amount of ice (190 Gt) broadly comparable with Totten
Glacier (236 Gt). There have also been several reports of inland thinning of Denman's fast-
flowing trunk (Flament and Remy, 2012; Helm et al., 2014; Young et al., 2015; Schröder et
al., 2018) and its grounding line has retreated over the past 20 years (Brancato et al., 2020).
However, unlike Totten and other large glaciers which drain marine basins in Antarctica, there
has been no detailed study analysing any changes in its calving cycle, velocity or ice tongue
structure. This study reports on a range of remote sensing observations from 1962 to 2018 and
then brings these observations together with numerical modelling to explore the possible



drivers of Denman's long-term behaviour. The following section outlines the methods (section
2) used to generate the remote sensing observations (section 3) and we then outline the
numerical modelling experiments (section 4) that were motivated by these observations,
followed by the discussion (section 5).

**2. Methods**
*2.1 Ice front and calving cycle reconstruction*
We use a combination of imagery from the ARGON (1962), Landsat-1 (1972-74), Landsat 4-
5 (1989-1991), RADARSAT (1997) and Landsat 7-8 (2000-2018) satellites to create a time
series of ice-front position change from 1962-2018. Suitable cloud-free Landsat imagery was
first selected using the Google Earth Engine Digitisation Tool (Lea, 2018). Changes in ice-
front position were calculated using the box method, which uses an open ended polygon to take
into account any uneven changes along the ice-front (Moon and Joughin, 2008). To supplement
the large gap in the satellite archive between 1974 and 1989 we use the RESURS KATE-200
space-acquired photography from September 1984. This imagery is hosted by the Australian
Antarctic Data Centre, and whilst we could not access the full resolution image, the preview
image was sufficient to determine the approximate location of the ice-front and confirm that a
major calving event took place shortly before the image was acquired (Fig. S1).

*2.2 Velocity*
Maps of glacier velocity between 1972 and 2002 were created using the COSI-Corr (CO-
registration of Optically Sensed Images and Correlation) feature-tracking software (Leprince
et al., 2007; Scherler et al., 2008). This requires pairs of cloud-free images where surface
features can be identified in both images. We found three suitable image pairs from the older
satellite data: Nov 1972 – Feb 1974, Feb 1989 – Nov 1989, and Nov 2001 – Dec 2002. We
used a window size of 128 x 128 pixels, before projecting velocities onto a WGS 84 grid at a
pixel spacing of 1 km.
To reduce noise, we removed all pixels where ice speed was greater than ±50% the MEaSUREs
ice velocity product (Rignot et al., 2011b), and all pixels where velocity was <250 m yr$^{-1}$. Errors
are estimated as the sum of the co-registration error (estimated at 1 pixel) and the error in



surface displacement (estimated at 0.5 pixels). This resulted in total errors ranging from 20 to
73 m yr$^{-1}$. Annual estimates of ice speed between 2005-2006 and 2016-2017 were taken from
the annual MEaSUREs mosaics (Mouginot et al., 2017). These products are available at a 1
km spatial resolution and are created from the stacking of multiple velocity fields from a variety
of sensors between July and June in the following year. To produce the ice speed time-series,
we extracted the mean value of all pixels within a defined box 10 km behind Denman's
grounding line (see Fig. 3). To eliminate any potential bias from missing pixels, we placed
boxes in locations where all pixels were present at each time step.
We also estimated changes in the rate of ice-front advance between 1962 and 2018. This is
possible because inspection of the imagery reveals that there has been only one major calving
event at Denman during this time period because the shape of its ice front remained largely
unchanged throughout the observation period. Similar methods have been used elsewhere on
ice shelves which have stable ice fronts e.g. Cook East Ice Shelf (Miles et al., 2018). This has
the benefit of acting as an independent cross-check on velocities close to the front of the ice
tongue that were derived from feature tracking. Whilst the accuracy of velocity fields produced
by modern satellites (e.g. Sentinel-1; Landsat-8) have been established, the accuracy of velocity
fields produced by earlier, coarser resolution satellite images (e.g. Landsat-1) are largely
untested. The ice-front advance rate was calculated by dividing ice-front position change by
the number of days between image pairs. Errors are associated with co-registration (1 pixel)
and manual mapping of the ice-front (0.5 pixels), resulting in errors ranging from 6 to 73 m yr$^{-}$
$^{1}$. The general pattern of ice-front advance rates through time is in close agreement with feature
tracking-derived changes in velocity over the same time period.

## 3. Results

### 3.1 Ice tongue calving cycles and structure

Throughout our observational record (1962 - 2018) Denman Glacier underwent only one major
calving event, in 1984, which resulted in the formation of a large 54 km long (1,800 km$^2$)
tabular iceberg (Fig. 2a-c). Since this calving event in 1984 the ice-front has re-advanced 60
km and there have been no further major calving events (Fig. 2a, b), as indicated by minimal
changes to the geometry of its 35 km wide ice front. As of November 2018, Denman Glacier's
ice-front was approximately 6 km further advanced than its estimated calving front position
immediately prior to the major calving event in 1984 (Fig. 2a, b). However, given the absence



of any significant rifting or structural damage, a calving event in the next few years is unlikely.
This suggests the next calving event at Denman will take place from a substantially more
advanced position (>10 km) than its last observed event in 1984.
Following the production of the large tabular iceberg from Denman Glacier in 1984, it drifted
~60 km northwards before grounding on the sea floor (Fig. 2c), and remained near stationary
for 20 years before breaking up and dispersing in 2004. Historical observations of sporadic
appearances of a large tabular iceberg in this location in 1840 (Cassin and Wilkes, 1858) and
1914 (Mawson, 1915), but not in 1931 (Mawson, 1932), suggest that these low-frequency,
high-magnitude calving events are typical of the long-term behaviour of Denman Glacier. In
1962, our observations indicate a similar large tabular iceberg was present at the same location
(Fig. 2d) and, through extrapolation of the ice-front advance rate between 1962 and 1974 (Fig.
2a), we estimate that this iceberg was produced at some point in the mid-1940s. However, the
iceberg observed in 1962 (~2,700 km$^2$) was approximately 50% larger in area than the iceberg
produced in 1984 (~1,700 km$^2$), and 35% longer (73 km versus 54 km). Thus, whilst Denman's
next calving event will take place from a substantially more advanced position than it did in
1984, it may not be unusual in the context of the longer-term behaviour of Denman Glacier
(Fig. 2a).
There are clear differences in the structure of Denman Glacier between successive calving
cycles. In all available satellite imagery between the 1940s and the calving event in 1984 (e.g.
1962, 1972 and 1974) an increasing number of rifts (labelled R1 to R7) were observed on its
ice tongue throughout this time (Fig. 2e). The rifts periodically form ~10 km inland of
Chugunov Island (Fig. 2e), on the western section of the ice tongue, before being advected
down-flow. An analysis of the rifting pattern in 1974 and the iceberg formed in 1984 indicates
that the iceberg calved from R7 (Fig. 2c, e). In contrast, on both the grounded iceberg observed
in 1962 (Fid. 2d), which likely calved in the 1940s, and on the present day calving cycle (1984-
present; Fig. 3f), similar rifting patterns are not observed.

### 3.2 Ice Speed

We observed widespread increases in ice speed across the entire Denman system between
1972-74 and 2016-17, with accelerations of 19 ±5% up to 50 km inland of the grounding line
along the main trunk of the glacier (Fig. 3a). Specifically, at box *D*, 10 km inland of the
grounding line, ice flow speed increased by 17 ±4% between 1972-74 and 2016-17 (Fig. 3c).





The largest rates of acceleration at box *D* took place between 1972-74 and 1989 when there
was a speed-up of 11 ±5%. Between 1989 and 2016-17 there was a comparatively slower
acceleration of 3 ±2% (Fig. 3c). The advance rate of the ice-front followed a similar pattern,
but accelerated at a much greater rate. The ice-front advance rate increased by 26 ±5% between
1972-74 and 1989, whilst increasing at a slower rate between 1989 and 2018 (9 ±1%; Fig. 3b).
At box *S* on the neighbouring Scott Glacier, we observed a 17 ±10% decrease in velocity
between 1972-74 and 2016-17 (Fig. 3d). Similar decreases in ice flow speed are also observed
near the shear margin between Shackleton Ice Shelf and Denman Glacier (Fig. 3a, e). The net
result of an increase in velocity at Denman Glacier and decreases in velocity either side at the
Shackleton Ice Shelf and Scott Glacier is a steepening of the velocity gradient at the shear
margins (Fig. 3e). Ice speed profiles across Denman Glacier also indicate lateral migration of
the shear margins of ~5 km in both the east and west directions through time (Fig. 3e).

*3.3 Lateral migration of Denman's ice tongue*
A comparison of satellite imagery between 1974 and 2002, when Denman's ice-front was in a
similar location (e.g. Fig. 4b, c), reveals a lateral migration of its ice tongue and a change in
the characteristics of the shear margins. North of Chugunov Island, towards the ice-front, we
observe a bending and westward migration of the ice tongue in 2002, compared to its 1974
position (Fig. 4b, c). In 1974, the ice tongue was intensely shearing against Chugunov Island,
as indicated by the heavily damaged shear margins (Fig. 4d). However, by 2002 the ice tongue
made substantially less contact with Chugunov Island because this section of the ice tongue
migrated westwards (Fig. 4d, e). South of Chugunov Island there was a greater divergence of
flow between the Denman and Scott Glaciers in 2002 compared to 1974, resulting in a more
damaged shear margin (Fig. 4d, e). On the western shear margin between Shackleton Ice Shelf
and Denman's ice tongue there was no obvious change in structure between 1974 and 2002
(Fig. 4f, g). However, velocity profiles in this region show an eastward migration of the fast
flowing ice tongue (Fig. 3e).

**4. Numerical Modelling**
*4.1. Model Set-Up and Experimental Design*



To help assess the possible causes of the acceleration of Denman Glacier since 1972 and the
importance of changes we observe on Denman's ice tongue, we conduct diagnostic numerical
modelling experiments using the finite-element, ice dynamics model Úa (Gudmundsson et al,
2012). Úa is used to solve the equations of the shallow-ice stream or `shelfy-stream'
approximation, (SSA , Cuffey & Paterson, 2010). Previously the model has been used to
understand rates and patterns of grounding line migration, and glacier responses to ice shelf
buttressing and ice shelf thickness (e.g. Reese et al., 2018; Hill et al., 2019; Gudmundsson et
al., 2019), and has been involved in several model intercomparison experiments (e.g. Pattyn et
al., 2008; 2012; Leverman et al., 2020).
Modelled ice velocities are calculated on a finite-element grid using a vertically-integrated
form of the momentum equations. The model domain consists of 93,371 elements with
horizontal dimensions ranging from 250 m near the grounding line to 10 km further inland.
Zero flow conditions are applied along the inland boundaries, chosen to match zero flow
contours from observations. Ice rheology is assumed to follow Glen's flow law, using stress
exponent $n$=3 and basal sliding is assumed to follow Weertman's sliding law, with its own
stress exponent, $m$ =3. Other modelling parameters related to ice rheology and basal conditions
are the basal slipperiness, $C$, and the rate factor, $A$. We initialized the ice-flow model by
changing both the ice rate factor $A$ (Fig. S2b) and basal slipperiness C (Fig. S2c), using an
inverse approach (Vogel, 2002), iterating until the surface velocities of the numerical model
closely matched the 2009 measurements of ice flow (Fig. S2a).

### *4.2. Perturbation Experiments*

We start from a baseline set-up at a fixed point in time where both velocity and ice geometry
are well-known. We chose 2009 for this baseline setup, because the calving front is in
approximately the same position as in 1972 when our glacier observations start. We use the
BedMachine (Morlighem et al., 2020) ice thickness, bathymetry and grounding line position
and MEaSUREs ice velocities for 2009 (Mouginot et al., 2017) as inputs. The baseline
simulation is then perturbed to test its response to a series of potential drivers that may be
responsible for the observed changes in ice geometry since the 1970s. Specifically, we apply
observation-based perturbations to test Denman's response to ice shelf thinning (i), grounding
line retreat (ii) and the unpinning of Denman's ice tongue from Chugunov Island (iii), which
are detailed below:



i.     To represent ice shelf thinning since 1972, we take the mean annual rate of ice-thickness

change from the 1994–2012 ice-shelf thickness change dataset (Paolo et. al., 2015). This annual

rate is then applied over the 37 years between 1972 and 2009 to obtain an estimate of the 1972

thickness distribution of the Shackleton Ice Shelf, Denman ice tongue and Scott Glacier. We

refer to this perturbation as 'ice shelf thinning' because the majority of the floating portions of

Denman's ice tongue and Shackleton Ice Shelf have thinned since 1994, although some

sections of Scott Glacier have actually thickened near its calving front (Fig. S3).

ii.    To represent grounding line retreat since 1972 we advanced Denman's grounding line

from its position in the 2009 baseline set-up by 10 km to a possible 1972 position (Fig. S3).

We justify a 10 km retreat since 1972 based on the rate of grounding-line retreat observed

between 1996 and 2017 (~5km; Brancato et al., 2020). This can be represented in the model

by either thickening the ice to ground it, or raising the bed into contact with the bottom of the

ice shelf. We have chosen to artificially raise the bedrock to the same height as the bottom of

the ice as the less disruptive method, as changing ice thickness would have an effect on ice

velocity in addition to that caused by moving the grounding line position. For the newly

grounded area, values of the bed slipperiness, C, are not generated in our model inversion, we

therefore prescribe nearest-neighbour values to those at the grounding line in the model

inversion.

iii.   To represent the pinning of Denman's ice tongue against Chugunov Island in the 1972

observations (e.g. Fig. 4d, e), we artificially raise a small area of bedrock on the edge of

Chugunov Island (Fig. S3). Bed slipperiness was set to a value comparable to that immediately

upstream of the grounding line.

These three adjustments are applied, both individually and in combination with each other, to

the baseline model setup to produce seven different simulations (E1-7) which perturb,

respectively:

E1.    Ice shelf thinning.
E2.    Grounding line retreat
E3.    Ice shelf thinning and grounding line retreat
E4.    Unpinning from Chugunov Island
E5.    Ice shelf thinning and unpinning from Chugunov Island



E6.     Grounding line retreat and unpinning from Chugunov Island
E7.     Ice shelf thinning, grounding line retreat and the unpinning from Chugunov Island
Below we compare the instantaneous change in ice velocity arising from each perturbation
experiment to observed changes in velocity, and then use these comparisons to understand the
relative importance of each process in contributing to Denman's behaviour over the past 50
years.

### 258    *4.3. Model results*

We show observed 2009 ice speed relative to each simulation which represent possible 1972
ice geometries (E1-7, Fig. 5b-h). In all cases, positive (red) values indicate areas where ice was
flowing faster and negative (blue) values show areas where ice was flowing slower in 2009
relative to each 1972 simulation. Perturbing ice shelf thickness to represent ice shelf thinning
since the 1970s results in higher velocities over both the grounded and floating portions of the
Denman system (E1, Fig. 5b). However, the simulated acceleration on Denman's ice tongue
(E1, Fig. 5b) is much larger than the observed acceleration, with the simulation showing a 50%
acceleration in the area just downstream from the grounding line compared to the observed
20% acceleration between 1972 and 2009 (E1, Fig. 5a). Thus, it would appear that ice shelf
thinning alone, is not consistent with the observed velocity changes on the Denman system.
Perturbing the grounding line to account for a possible grounding line retreat since 1972
simulates comparable changes in ice flow speeds to observations near Denman's grounding
line (E2, Fig. 5c), but it is unable to reproduce the observed increases in ice speed across
Denman's ice tongue (E2, Fig. 5c). Thus, grounding line retreat, alone, is also unable to
reproduce the observed pattern of velocity changes. Ice shelf thinning and retreating the
grounding line results in very similar patterns in ice speed change (E3, Fig. 5d) to that of the
grounding line retreat perturbation experiment (E2).
In isolation, simulating the unpinning of Denman's ice tongue from Chugunov Island has a
very limited effect on ice flow speeds, with no change in speed near the grounding line and a
very spatially limited change on the ice tongue (E4; Fig. 5e). However, when combining the
unpinning perturbation with either ice shelf thinning (E5; Fig. 5f) or grounding line retreat (E6;
Fig. 5g), it is clear that the unpinning from Chugunov Island causes an acceleration across
Denman's ice tongue. For experiment 5 this results in an even larger overestimate of ice speed
change across Denman's ice tongue in comparison to experiment 1, which only perturbs ice
shelf thickness. However, for experiment 6 the additional inclusion of the unpinning from
Chugunov Island to grounding line retreat results in a simulated pattern of ice flow speed
change very similar to observations. Specifically the unpinning from Chugunov Island has
caused an acceleration across the ice tongue that was not present in experiment 2. Combining
all three perturbations (E7, Fig. 5h) produces changes in ice velocity that are most comparable
to observations. Both the spatial pattern in ice speed change and the simulated ice speed within
box *D* (Fig. 5i) are very similar to observations for both experiments, and the enhanced
westward bending of the directional component of ice velocity in experiment E7 is more
consistent with the observed westward bending of the ice tongue (e.g. Fig. 2b).

**5. Discussion**
*5.1 Variation in Denman Glacier's calving cycle*
Our calving cycle reconstruction, combined with historical observations (Cassin and Wilkes,
1858; Mawson, 1914; 1932) hint that Denman's multi-decadal high-magnitude calving cycle
has remained broadly similar over the past 200 years. It periodically produces a large tabular
iceberg, which then drifts ~60 km northwards before grounding on an offshore ridge, and
typically remains in place for around 20 years before disintegrating/dispersing. However, more
detailed observations and reconstructions of its past three calving events have shown that there
are clear differences in both the size of icebergs produced and in ice tongue structure through
time (Fig. 2). The large variation (50%) in both the size of iceberg produced and the location
the ice front calved from indicates variability in its calving cycle.
Extending observational records for ice shelves that calve at irregular intervals, sizes or
locations is especially important because it helps to distinguish between changes in glacier
dynamics caused by longer-term variations in its calving cycle, and changes in glacier
dynamics forced by climate. For example, there have been large variations in ice flow speed at
the Brunt Ice Shelf over the past 50 years (Gudmundsson et al., 2017), but these large variations
can be explained by internal processes following interactions with local pinning points during
the ice shelf's calving cycle (Gudmundsson et al., 2017). In contrast, the widespread
acceleration of outlet glaciers in the Amundsen Sea sector (Mouginot et al., 2014) is linked to
enhanced intrusions of warm ocean water increasing basal melt rates (e.g. (Thoma et al., 2008;
Jenkins et al., 2018), leading to ice shelf thinning (Paolo et al., 2015) and grounding line retreat



(Rignot et al., 2011a). Thus, in the following section we discuss whether the observed speed-
up of Denman since the 1970s (Fig. 3) is more closely linked to variations in its calving cycle
(e.g. Brunt Ice Shelf) or if it has been driven by climate and ocean forcing (e.g. Amundsen
Sea).

### 5.2 What has caused Denman Glacier's acceleration since the 1970s?

We observe a spatially widespread acceleration of both Denman's floating and grounded ice.
This is characterised by a 17 ±4% increase in ice flow speed near the grounding line between
1972 and 2017 (Fig. 3c) and a 36 ±5% acceleration in ice-front advance rate from 1972-2017,
or 30 ±5% increase in ice-front advance rate between 1962 and 2017 (Fig. 3b). Our estimates
of the acceleration in ice front advance rate are of a comparable magnitude to the 36%
acceleration of the ice tongue between 1957 and 2017, based on averaged point estimates across
the ice tongue from repeat aerial surveys (Dolgushin, 1966; Rignot et al., 2019). Taken
together, this suggests a limited change in ice tongue speed between 1957 and 1972, before a
rapid acceleration between 1972 and 2017. However, the rate of acceleration throughout this
period has not been constant (Fig. 3b, c). From 1972 to 1990, observations indicate that ice
accelerated approximately three times faster on the ice tongue (Fig. 3b) and twice as fast at the
grounding line (Fig. 3c) in comparison to accelerations in these areas between 1990-2017.
When comparing these observations against our numerical modelling experiments we find that
a combination of grounding line retreat, changes in ice shelf thickness and the unpinning of ice
from Chugunov Island (Fig. 5h) are all required to explain an acceleration of a comparable
magnitude and spatial pattern across the Denman system.
Averaged basal melt rates across the Shackleton/Denman system are comparable to the Getz
Ice Shelf (Depoorter et al., 2013; Rignot et al., 2013). Close to Denman's deep grounding line,
melt rates have been estimated at 45 m yr$^{-1}$ (Brancato et al., 2020), suggesting the presence of
modified Circumpolar Deep Water in the ice shelf cavity. At nearby Totten Glacier (Fig. 1a),
wind-driven periodic intrusions of warm water flood the continental shelf and cause increased
basal melt rates (Rintoul et al., 2016; Greene et al., 2017) and grounding line retreat (Li et al.,
2015). It is possible that a similar process may be responsible for some of the observed changes
at Denman Glacier. Hydrographic data collected from the Marine Mammals Exploring the
Oceans Pole to Pole consortium (Treasure et al., 2017) show water temperatures of -1.31 to -
0.26 °C at depths between 550 and 850 m on the continental shelf in front of Denman (Brancato



et al., 2020). Thus, whilst not confirmed, there is clear potential for warm water to reach
Denman's grounding zone and enhance melt rates.
Recent observations of grounding line migration at Denman have shown a 5 km retreat along
its western flank between 1996 and 2017 (Brancato et al., 2020). However, over this time
period there was a limited change in the speed of Denman (2001-2017; 3 ±2% acceleration;
Fig. 3c) and our time series indicates that the acceleration initiated earlier, at some point
between 1972 and 1990 (Fig. 3c). Reconstructions of the bed topography near the grounding
line of Denman Glacier show that the western flank of Denman's grounding line was resting
on a retrograde slope in 1996, a few kilometres behind a topographic ridge (Brancato et al.,
2020). One possibility is that Denman's grounding line retreat initiated much earlier at some
point in the 1970s in response to increased ocean temperatures enhancing melting of the ice
tongue base. This initial grounding line retreat and possible ocean-induced ice tongue thinning
may have caused the initial rapid acceleration between 1972 and 1990, before continuing at a
slower rate. However, our numerical modelling shows that whilst the combination of the retreat
of Denman's grounding line and ice tongue thinning can produce a similar magnitude of
acceleration near the grounding line to observations (E3; Fig. 5e), these modelled processes
cannot explain the widespread acceleration across the ice tongue (e.g. E4; Fig. 5d).
In order to simulate a comparable spatial acceleration across both Denman's grounded and
floating ice to observations, the un-pinning of ice from Chugunov Island following Denman's
last calving event in 1984 is required (e.g. E6 & 7; Fig. 5g, 5h). In isolation, the reduction in
contact with Chugunov Island has had no effect on ice flow speeds at both Denman's grounding
line and ice tongue (E4; Fig. 5d). However, when combined with grounding line retreat and ice
tongue thinning, the spatial pattern of simulated ice speed change across the ice tongue more
closely resemble observations (E6 & 7; Fig. 5g, 5h). Specifically, the unpinning of the ice
tongue from Chugunov Island has caused an acceleration across much of Denman's ice tongue.
The most likely explanation as to why the unpinning from Chugunov Island only influences
ice speed patterns in combination with ice tongue thinning and grounding line retreat, and not
in isolation, is that ice tongue thinning and grounding line retreat have caused a change in the
direction of flow of the ice tongue since the 1970s. In all simulations that perturb either ice
tongue thickness or retreat the grounding line (Fig. 5b, c, e, f, g, h), there is a clear westward
bending in ice flow direction near Chugunov Island which results in a reduction in contact
between the ice tongue and Chugunov Island. This is consistent with observations that show a
distinctive westward bending of Denman's ice tongue since the 1970s (Fig. 2b). These findings



therefore suggest that the reduction in contact with Chugunov Island following Denman's calving event in 1984 caused an instantaneous acceleration across large sections of its ice tongue, meaning that this calving event has had a direct impact on the spatial pattern of acceleration observed between 1972 and 2017. However, because of the westward bending of Denman's ice tongue during its re-advance following its 1984 calving event, the ice tongue now makes limited contact with Chugunov Island (e.g. Fig. 4e) and has a very limited effect on ice flow speeds (e.g. E4; Fig. 5e ).

The acceleration of Denman's ice tongue following its last major calving event in 1984 may have also caused a series of positive feedbacks resulting in further acceleration. We observe a steepening of the velocity gradient across Denman's shear margins and a pattern of the acceleration of the dominant Denman ice tongue and slowdown of the neighbouring Shackleton Ice Shelf and Scott Glacier (Fig. 3a). We also observe the lateral migration of the shear margins at sub-decadal timescales (Fig. 3e). These distinctive patterns in ice speed change are very similar to those reported at the Stamcomb-Wills Ice Shelf (Humbert et al., 2009) and between the Thwaites Ice Tongue and Eastern Ice Shelf (Mouginot et al., 2014; Miles et al., 2020), and are symptomatic of a weakening of shear margins. Therefore, we suggest that at Denman, after the initial acceleration following the reduction in contact with Chugunov Island, the shear margins may have weakened causing further acceleration. We do not include this process in our numerical experiments, and it may explain the divergence between observations and simulated ice speed change in the neighbouring Shackleton Ice Shelf and Scott Glacier (Fig. 3a; Fig. 5).

Overall, our observations and numerical simulations suggest that the cause of Denman's acceleration since the 1970s is complex and likely reflects a combination of processes linked to the ocean and a reconfiguration of Denman's ice tongue. One possibility is that the acceleration of ice across Denman's grounding line has almost entirely been driven by warm ocean forcing driving grounding line retreat and ice tongue thinning, with the unpinning of Denman's ice tongue from Chugunov Island only causing a localised acceleration across floating ice. An alternative explanation is that warm ocean forcing has caused ice tongue thinning and grounding line retreat, but the acceleration behind the grounding line has been enhanced through time by changes in ice tongue configuration. Either way, our results highlight that both oceanic processes and the changes in ice tongue structure associated with Denman's calving event have been important in causing Denman's observed acceleration.






### 5.3 Future evolution of Denman Glacier


In the short-term, an important factor in the evolution of the wider Denman/Shackleton system
is Denman's next calving event. Whilst our observations do not suggest that a calving event is
imminent (next 1-2 years), our calving cycle reconstruction indicates that a calving event at
some point in the 2020s is highly likely. Because the calving cycle of Denman Glacier has
demonstrated some variability in the past (e.g. Fig. 2), the precise geometry of its ice tongue
after this calving event cannot be accurately predicted. In particular, it is unclear how
Denman's ice tongue will realign in relation to Chugunov Island following its next calving
event. For example, if following Denman's next calving event the direction of ice flow shifts
eastwards to a similar configuration to the 1970s and the ice tongue makes contact with
Chugunov Island, the increased resistance could slowdown Denman's ice tongue for the
duration of its calving cycle, but it is unclear if any slowdown could propagate to the grounding
line. Thus, this event may have important implications for the evolution of the
Denman/Shackleton system for multiple decades.
In the medium-term (next 50 years) atmospheric warming could also have a direct impact on
the stability of the Denman/Shackleton system. Following the collapse of Larsen B in 2002,
Shackleton is now the most northerly major ice shelf remaining in Antarctica, with most of the
ice shelf lying outside the Antarctic Circle. Numerous surface meltwater features have been
repeatedly reported on its surface (Kingslake et al., 2017; Stokes et al., 2019; Arthur et al.,
2020). There is no evidence that these features currently have a detrimental impact on its
stability, but there is a possibility that projected increases in surface melt (Trusel et al., 2015)
could increase the ice shelves vulnerability to meltwater-induced hydrofracturing

### 6. Conclusion


We have reconstructed Denman Glacier's calving cycle to show that its previous two calving
events (~1940s and 1984) have varied in size by 50% and there have been clear differences in
ice tongue structure, with a notable unpinning from Chugunov Island following the 1984
calving event. We also observe a long-term acceleration of Denman Glacier across both
grounded and floating sections of ice, with both the ice front advance rate and ice near the
grounding line accelerating by 36 ±5% and 17 ±4%, respectively, between 1972 and 2017. We





show that in order to simulate a post-1972 acceleration that is comparable with observations,
its grounding line must have retreated since the 1970s. We also highlight the importance of the
reconfiguration of the Denman ice tongue system in determining the spatial pattern of
acceleration observed.
The recent changes in the Denman system are important because Denman's grounding line
currently rests on a retrograde slope which extends 50 km into its basin (Morlighem et al.,
2019; Brancato et al., 2020), suggesting clear potential for marine ice sheet instability. Given
the large catchment size, it has potential to make globally significant contributions to mean sea
level rise in the coming decades (1.49 m; Morlighem et al., 2020). Crucial to assessing the
magnitude of any future sea level contributions is improving our understanding of regional
oceanography, and determining whether the observed changes at Denman are the consequence
of a longer-term ocean warming. This is in addition to monitoring and understanding the
potential impact of any future changes in the complex Shackleton/Denman ice shelf system.
In a wider context our results add to the growing body of evidence that some major East
Antarctic outlet glaciers, with multi-meter sea-level equivalent catchments have responded to
changes in ocean-climate forcing over the past 100 years and, therefore, will be sensitive to
projected future warming. The Ninnis Ice Tongue has retreated (Frezzotti et al., 1998), part of
the Cook Ice Shelf has collapsed (Miles et al., 2018) and Totten's grounding line is retreating
(Li et al., 2015), and it has been losing mass for several decades (Rignot et al., 2019). However,
despite recent improvements (e.g. Morlighem et al., 2020), our understanding of bed
topography, oceanography, bathymetry and ice shelf cavity geometry in these key regions of
East Antarctic still lag behind that of other regions e.g. Amundsen Sea sector, making the
accurate assessment of the magnitude and rates of future sea level contributions challenging.

**Acknowledgements**
This work was funded by the Natural Environment Research Council (grant number:
NE/R000824/1). Landsat and the declassified historical imagery from 1962 is freely available
and can be downloaded via Earth Explorer (https://earthexplorer.usgs.gov/). COSI-Corr is
available at http://www.tectonics.caltech.edu/slip_history/spot_coseis/download_software
.html. The source code for Úa is available at https://doi.org/10.5281/zenodo.3706624.
MEaSUREs     annual     ice     velocity     maps     are     available     at
https://doi.org/10.5067/9T4EPQXTJYW9. We also thank Eric Rignot for providing digitized



estimates of ice flow speed across parts of Denman's ice tongue, based on the mapped estimates
of Dolgushin et al. (1966).

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

**Figures**

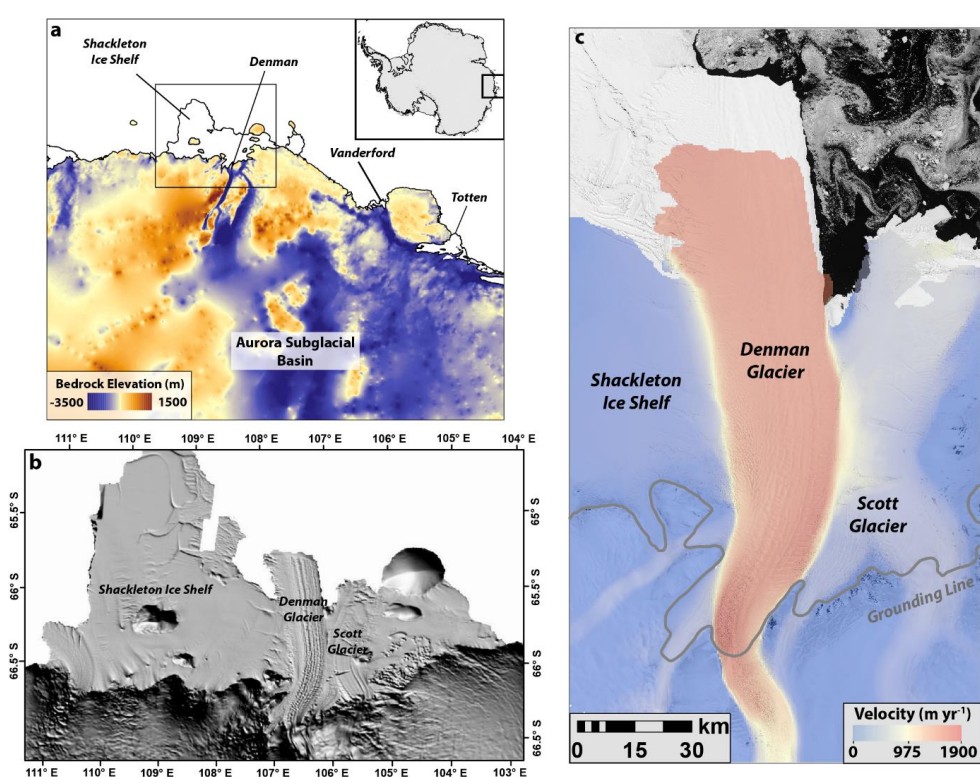

**Figure 1: a)** Bedrock elevation of the Aurora Subglacial Basin (Morlighem et al., 2020). Note
the deep trough inland of the Denman grounding line. **b)** REMA hill-shade DEM of Denman
Glacier, Scott Glacier, and the Shackleton Ice Shelf (Howat et al., 2019). Note the complex
composition of the Shackleton Ice Shelf with several pinning points. **c)** MEaSUREs velocity
of Denman Glacier (Rignot et al., 2011) overlain on a Landsat-8 image from November 2018.
Note the steep velocity gradient between the Shackleton Ice Shelf, Denman Glacier and Scott
Glacier.

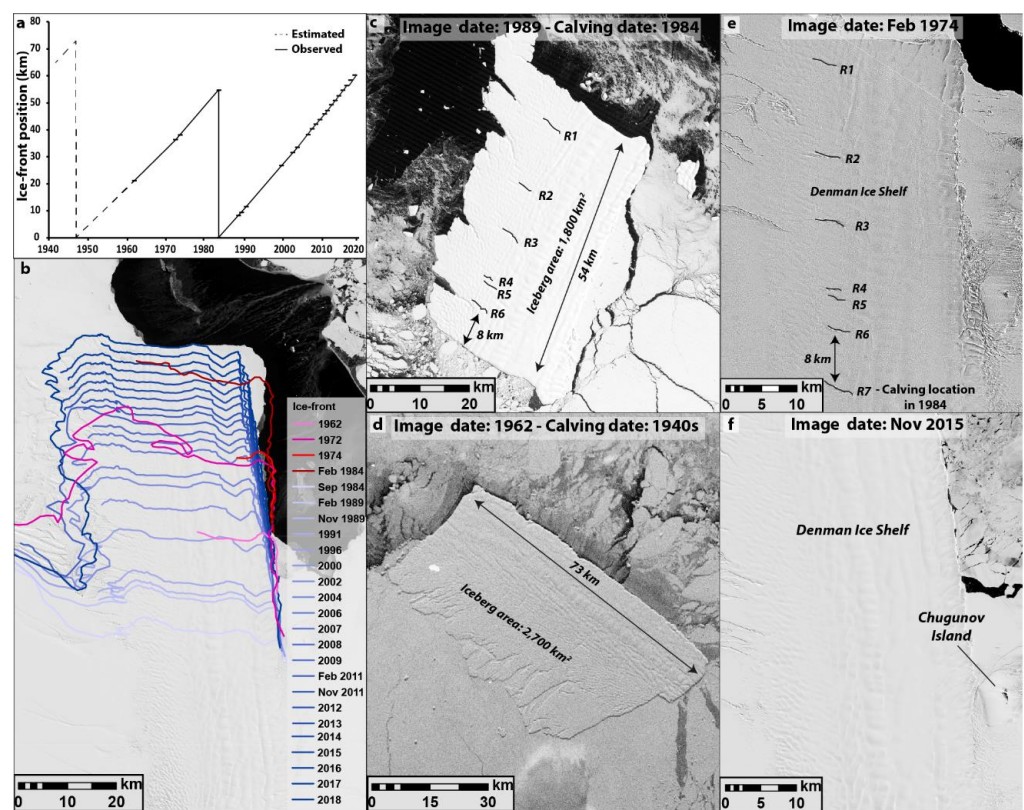




**Figure 2: a)** Reconstructed calving cycle of Denman Glacier 1940-2018. **b)** Examples of ice-
front mapping 1962-2018. Note the change in angle of the ice shelf between its present (light
blue – dark blue lines) and previous (pink-red lines) calving cycle. **c)** and **d)** Images of the
grounded iceberg produced by Denman in 1984 (c) and in the 1940s (d). **e)** and **f)** Difference
in ice shelf morphology between 1974 and 2015. Note the presence of rifting in **e** (digitized in
black).


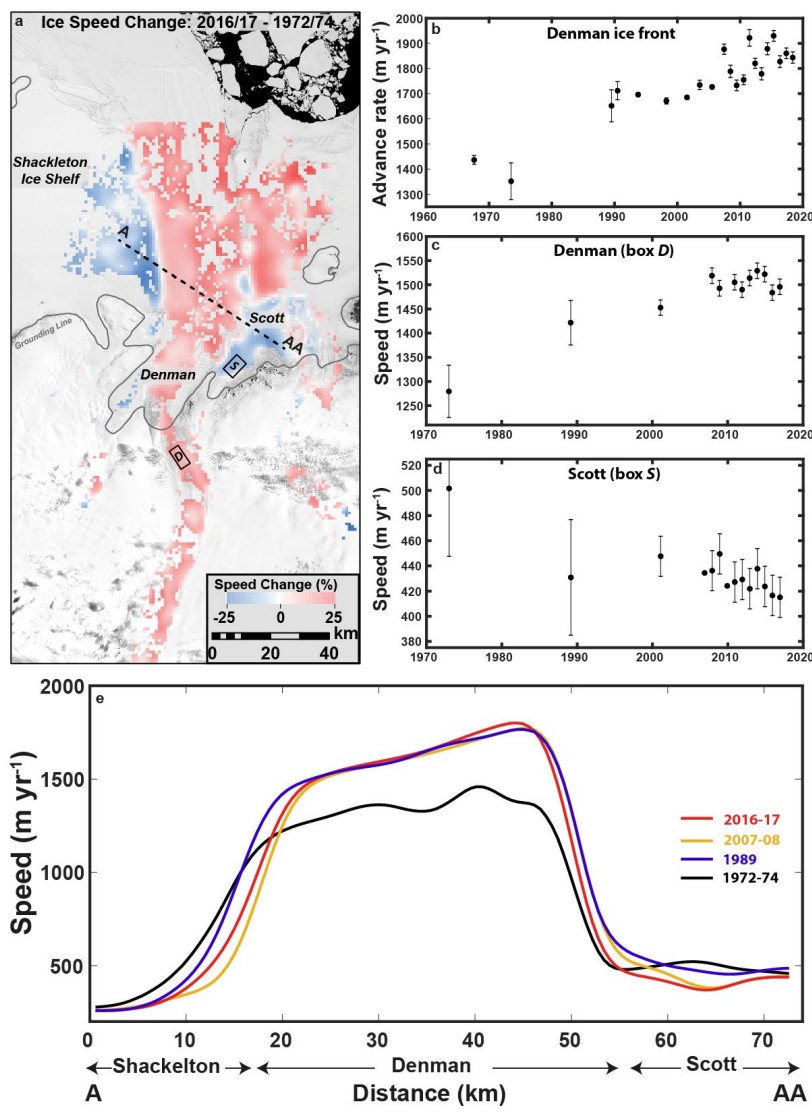

**Figure 3: a)** Percentage difference in ice speed between 2016-17 and 1972-74. Red indicates a relative in increase in 2016-17 and blue a relative decrease in 2016-17. **b)** Time series of the advance rate of the Denman ice-front 1962-2018. **c)** Time series of mean ice speed from box *D*, 1972-2017) approximately 10 km behind the Denman grounding line. **d)** Time series of mean ice speed from box *S*, on Scott Glacier, 1972-2017. **e)** Ice speed profiles across the Shackleton-Denman-Scott system from 1972-74, 1989, 2007-08 and 2016-17.





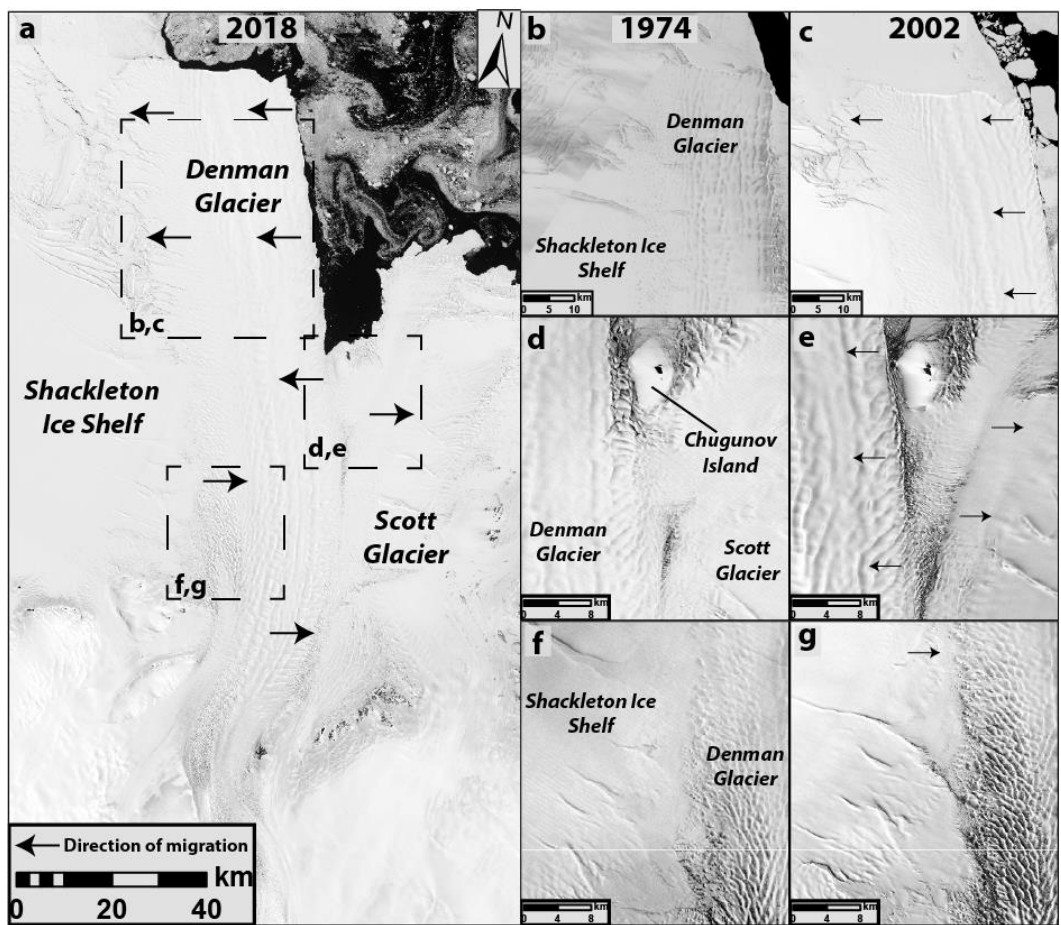

**Figure 4: a)** Direction of ice tongue migration since 1974. **b), d) and f)** Close-up in examples of ice tongue structure and location in 1974. **c), e) and g)** Close-up in examples of ice tongue structure and position in 2002. In particular, note the reduction in contact between Denman Glacier and Chugunov Island between 1974 (d) and 2002 (e).

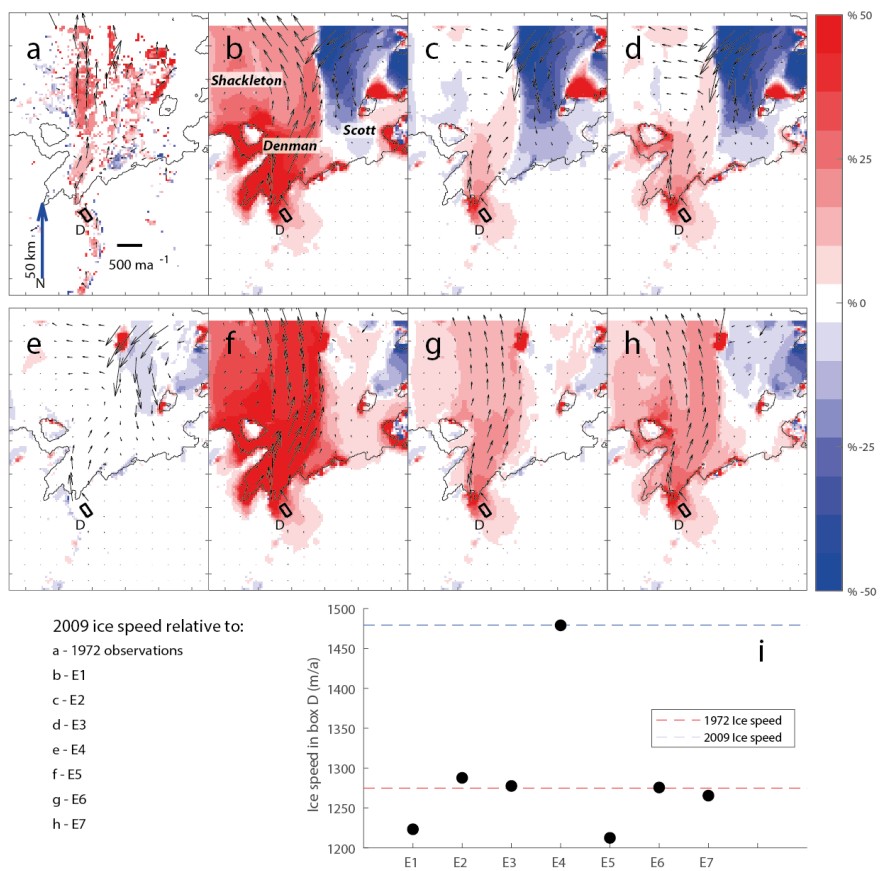

721

**Figure 5 – The effect of varying ice geometry on ice flow:** Ice velocity difference between
2009 observations and **a)** observations from 1972, and **b)-h)** seven experiments which perturb
2009 ice geometry to represent possible 1972 ice geometry configurations. The seven
experiments are: **b)** ice shelf thinning (E1), **c)** grounding line retreat (E2), **d)** ice shelf thinning
and grounding line retreat (E3), **e)** unpinning from Chugunov Island (E4), **f)** ice shelf thinning
and unpinning from Chugunov Island (E5), **g)** grounding line retreat and unpinning from
Chugunov Island (E6) and **h)** combining ice shelf thinning, grounding line retreat and the
unpinning from Chugunov Island (E7). Note that red indicates areas where ice is flowing faster
in 2009 and blue indicates areas that are flowing slower with arrows showing the direction and
magnitude of change when compared to the 1972 perturbations. Finally, **i)** the mean speed in
box D (located just upstream of the grounding line, shown in black) in the perturbed (1972)
configuration, with the observed 1972 and 2009 speed shown in red and blue, respectively. E7
most closely matches the speed observed in box D, the spatial pattern of the observed
acceleration and the westward bending of Denman's ice tongue.