# Peer review of "Recent acceleration of Denman Glacier (1972-2017), East Antarctica, driven by grounding line retreat and changes in ice tongue configuration."

_The Cryosphere, 2020_

## Referee Comment (RC1) · Chad Greene (Referee) · 30 Jul 2020

**Recent acceleration of Denman Glacier (1972-1 2017), East Antarctica, driven by grounding line retreat and changes in ice tongue configuration.**

*Miles et al.*

Review by Chad A. Greene
July 29, 2020

**General comments**

In this paper, Miles et al. generate observations of velocity and calving front positions of Denman Glacier, then they apply various perturbations to the geometry of a simple ice sheet model to determine what mechanisms might explain the observed behavior of the ice system. The study is elegantly designed and the manuscript is very well written.

The historical context provided by the ARGON and other early satellite photography is valuable, and I appreciate the work the authors have done to sift through the archives, in which they found a coherent and story to tell. I especially appreciate that the authors present background information in a way that sets the stage for understanding why this research was performed and what the results might mean for the future. The paper is packed with little insights such as the fascinating link between ice shelf thinning and flow direction, yet despite the density of information the text flows effortlessly. It was an enjoyable read, I learned a bit, and I recommend the paper for publication after the data and code are made publicly available.

**Specific comments**

**Data and code sharing:** This is important work, and in the future there will undoubtedly be more studies of the flow speed of Denman Glacier. Part of that work will involve reporting on changes that will have occurred since the publication of this study, and there's a good chance the authors of such a future study will want to begin by plotting velocity profiles from the 2020s on top of the results shown in Fig 3e. To allow others to build on this work, please include the coordinates and measured velocities shown in Fig 3e as supplemental material to the manuscript or upload to a data repository such as PANGAEA. Similarly, the authors have generated a wonderful calving front extent dataset shown in Fig 2b...Please share it so others may build on this work! Same goes for the Úa model code that was used to generate these results—I would love to see it after reading this paper.

**L450-464:** At the end of a near-perfect manuscript in which each sentence brings new insights while setting the stage gracefully for the sentence that follows it, the final couple of paragraphs transition into a series of miscellaneous ideas that are related to, but not clearly relevant to the

main story of the manuscript. Each of these points could be expanded by a few sentences to help glue them to the findings of the study, but I don't think there's a need. Rather, most of the last two paragraphs could be deleted without detriment to the manuscript. I recommend simply reminding readers of the key historical behavior and/or future potential of Denman, and placing your results firmly in that context.

**Technical corrections**

**L26-27:** The general sentiment of the final sentence of the abstract is reasonably supported by the analysis presented in this manuscript, but the phrase "...over the coming century" constrains the prediction a bit too tightly, because timescales of ice response are not directly discussed in this paper.

**L63:** "...a range of remote sensing observations..." Make this sentence more clear by stating explicitly that velocity and ice front position observations are analyzed.

**L94-95:** Indicate how the 1 pixel and 0.5 pixel error estimates were obtained.

**L113-114:** Again, indicate how the error estimates were obtained.

**L155-156:** I don't think "accelerations" should be plural here. I recommend replacing "...with accelerations of 19±5%..." with "...with an overall acceleration of 19±5%..." Unless I've misunderstood the meaning of the sentence, in which case, please clarify.

**L246-252:** I had to make this table to keep the experiments straight in my head. It may save others the same trouble to have the experiments explicitly tabulated in the manuscript.

|    | ice shelf thinning | grounding line retreat | unpinning from Chugunov |
|----|:---:|:---:|:---:|
| E1 | ✓ |   |   |
| E2 |   | ✓ |   |
| E3 | ✓ | ✓ |   |
| E4 |   |   | ✓ |
| E5 | ✓ |   | ✓ |
| E6 |   | ✓ | ✓ |
| E7 | ✓ | ✓ | ✓ |

**L259:** I think "each simulation" should be "each of the seven simulations".

**L330-331:** Comparing accelerations as scalar multiples of each other is confusing, because I don't have any intuition for what it means if one thing has three times the acceleration of another thing. Actually, the sentence says "the ice accelerated approximately three times

faster" and if *a* is three times more than *b*, then *a*=4*b*, which makes the sentence even more confusing. Reword.

**L424:** Explicitly state what "this event" is. i.e., "Thus, the next major calving event…" And if the implications are important, don't just imply them—directly state what is implied are. i.e., "…could dictate the flow speed and direction of the…"

**L426:** The abstract and conclusion describes 50 years as "long term" but here 50 years is described as "medium term". I recommend removing these types of relative descriptions of timescales throughout.

**L433:** Period at the end of the sentence.

**L453:** I'm not sure what the word "This" refers to.

**Figure 1:** This figure shows bedrock topography, ice velocity, and the spatial distribution of pinning points in the Shackleton Ice Shelf. These are all valuable as context for the study, and I appreciate that the figure legend clearly states the important things that viewers should take note of.

My only complaint is that each variable is plotted in a separate panel, so understanding relationships between velocity, bed topography, and pinning points requires pinballing between all three panels as a way to mentally try to bring the variables all into one figure. Reconstituting the three variables is made more difficult by the fact that each panel shows different spatial extents, and at different scales.

I recommend experimenting with transparency, vectors, or contours to show all three variables on one plot. That would also allow more detail, as a single panel could be enlarged to fill the entire width of the page. For example, something like the following would be a way to show ice velocity in the context of surface features and the bed topography that ties Denman Glacier to the ASB:

[Figure]

Alternatively, the following semitransparent ice speed map overlaid on a MODIS MOA image shows the surface features you're trying to highlight, and I'm showing marine-based ice with a cobalt blue stippled region:

[Figure]

When I look at the figures above, I find myself drawn in, naturally following the direct path from the calving front of Denman Glacier, upstream through the sinuous, deep trough, and into the wide expanse of the marine-based area of the ASB. That's a very different experience from looking at multiple different panels shown at different scales and being tasked with mentally trying to put them all together.

Consider placing a box labeled "Fig. 2b" on Figure 1 to indicate the extents of Figure 2b.

**Figure 2:** As a logical sequence, I'd also put panel b before panel a, because currently panel b shows the direct observations and then panel a shows a quantified version of the observations.

I'm also having difficulty understanding where panels c-f are in relation to panel b. There are no recognizable reference points in any of the images, so it's difficult to place them in space. I would typically assume that the image orientation remains constant across all panels, but the spatial extent and even the spatial scale is different in each panel, so everything is in question.

It's also tempting to assume panels c-f depict a sequence of events, but they are presented out of chronological order, so there's an extra little bit of mental bookkeeping that viewers must do to reconstruct what has happened to this glacier tongue since 1962.

If it makes sense to do so, I'd like to see the spatial extents of panels c-f remain constant across all panels, so it will be easy to follow changes over time. I suspect the entire figure would be easier to digest if the panels were rearranged, and if the times of panels c-f were marked directly on the ice front position time series. Something along these lines feels much more intuitive to me:

[Figure]

Or perhaps the time series plot would fit best below the calving-front map, but however you do it, I think the sequence of the panels is important for understanding what story is being told by the figure, and drawing direct connections between all the panels (such as by labeling the times of panels c-f directly on the time series) will help viewers see how the information is all related.

Also, more can be done in the caption to help readers understand the connection between ice front position and ice morphology. This could be just a sentence or two, but just something to help viewers see why R1 through R7 are labeled.

**Figure 3:** State which grounding line dataset is being shown here. Partly to give credit to the data producers, but also because InSAR and break-in-slope grounding lines don't agree here, and knowing which GL is plotted would help readers visually identify where certain features are relative to a particular GL.

**Figure 5:** The mental ledger keeping required to interpret this figure is not terribly onerous, but it involves more steps than are necessary. For example, if I want to know what's being depicted in panel g, I must go to the legend in the bottom left, where I see g corresponds to E6, then I find E6 in panel i, and then I say, "okay, E6 falls closer to a dashed line than most of the other dots do." And then I get curious about the outlier dot corresponding to E4 on the x axis. "I wonder what that is," I think, and so I repeat the process backward, going to the legend in the lower left of the figure to find that E4 corresponds to panel e, so then I look at panel e and I see a mostly blank white panel. At no point in that process is there any indication of what any of

these letters and numbers mean, because the figure has been stripped of all links to physical processes.

I recommend eliminating the legend from the bottom left and simply labeling "E1: ice shelf thinning," "E2: grounding line retreat," etc., directly on panels a-h, either as titles outside the plot or in the empty space in the bottom of each panel. That would also free up the text of the figure caption to focus on physical processes, rather than bookkeeping.

In the text caption, hammer home the main point by stating that E7 most closely matches observed velocities, suggesting that ice shelf thinning, grounding line retreat, and unpinning from Chugunov Island have all occurred since 1972.

Panel c shows the effects of grounding line retreat, but grounding line retreat itself is not shown. It's hard to gauge spatial scales here, but would a 10 km retreat be visible at this scale? If so, show both the 1972 and 2009 grounding lines.

Panel e shows the effects of unpinning from Chugunov Island. It would be helpful to label Chugunov Island directly on that panel.

I appreciate that panel i puts most meaningful region of velocities of each experiment in context with each other, while also showing the observed 1972 and 2009 velocities, but the panel comes up short in communicating the main point. It's relatively innocuous, so keep the panel if it feels important, but know that it adds a layer of complication to interpreting the figure as a whole. If you'd like to keep it, I recommend including Box D velocities from panel a as a data point. That would make it more clear how E4 got so out of line relative to the others. If you decide to eliminate panel i, the ice speed values could simply be printed next to Box D in their respective panels and/or included as a column in the table I recommended above.

I find myself leaning in close and squinting to see the details around the grounding line. Then I zoom the pdf to 300% and realize the problem isn't my eyesight, but the coarse resolution of the graphics. I recommend recreating the figure at higher resolution (If it's Matlab, try `export_fig myfigure.png –r600` for 600 dpi) and enlarging the figure to fill the full width of the page so readers can see the beautiful details that are surely present in this data.

---

## Referee Comment (RC2) · Anonymous Referee #2 · 2 Aug 2020

**Review report for:**
**"Recent acceleration of Denman Glacier (1972-1 2017), East Antarctica, driven by grounding line retreat and changes in ice tongue configuration."**

**General comments**

In this manuscript by Miles et al., the authors explore the connection between surface ice velocity acceleration and calving events for Denman Glacier, East Antarctica. Particularly, to explore this connection, the authors apply several tools ranging from remote sensing observations to ice sheet modelling to grasp which mechanisms might explain the acceleration and ground line retreat of Denman Glacier.

The manuscript is very well written and flows quite smoothly in the description of the methods and the used ice sheet models. As a remote sensing expert, I really liked the contribution brought by the historical remote sensing data, which are generally very difficult to find.
The main hassle for me was going through all the figure. The absence of system coordinates makes very difficult to go from one figure to the other and make connections and comparisons between the figure. Therefore, I recommend the paper to be published after major revisions.
In the following, some additional comments.

1. Line 11, identifying Denman glacier as the largest is really vague. For a reader, it would be nice to specify that e.g. this glacier is the largest contributor to sea level rise in East Antarctica (after Totten Glacier)
2. One of the major statements of the paper is that to explain the acceleration pattern of Denman Glacier it is required to have a combination of grounding line retreat, changes in ice shelf thickness and unpinning of ice from Chugonov Island (lines 331-334). I am wondering if the unpinning of Chugunov Island comes from observations. Did the authors observe the unpinning in their data?
3. Line 94-95 and line 109-112. Have the authors quantitatively determined the uncertainty of surface ice velocity obtained from historical data? Instead of providing a ball park number, it would be nice to have a section in the supplementary describing how they quantified this uncertainty.
4. To my understanding, to perform their analysis the authors have used the Measure grounding line product for Denman. Do you know to which sensor and which year this grounding line belongs to? Also, have the authors tried to include more recent grounding line products?
5. I find really difficult to go through all the model experiments that the authors have performed. Is there a way to visually summarize them in the manuscript, e.g. using a table?
6. To explain the acceleration of Denman glacier, in Section 5.2 the authors discuss whether this is driven by warm water intrusion (e.g. ice-ocean interactions) or calving events. I find this section a bit confusing especially because it introduces a lot of elements which have not previously discussed in the manuscript. Do the authors simulate with their ice model

the effect of the ice-ocean interaction? Also, does the model take into account the effect of basal melt of Denman ice shelf?

7. As a general comment, ALL figures need to have coordinates grid. For me it was extremely difficult to move from one

8. I find the colorbar in figure 1-a to be confusing. Maybe all areas with an elevation over 0 should be marked with the same color, e.g. white.

9. Without coordinates is a bit difficult to link Fig.1b and Fig.1c. I would recommend to highlight the area in Fig.1c using a box in Fig.1b

10. Date of used Landsat images and acknowledgement for the used Landsat product should be included in the figure caption. Please, see USGS website for getting the proper data citation.

11. For Fig.2, each panel focus on a different area of the ice shelf. I would recommend to have a bigger picture of the ice shelf on the side and try to help the reader understand on which areas you are focusing/zooming on. Please, include latitude and longitude (or polar sterographic) coordinates for each figure.

---

## Referee Comment (RC3) · Anonymous Referee #3 · 22 Sep 2020

**Comments to "Recent acceleration of Denman Glacier (1972-2017), East Antarctica, driven by grounding line retreat and changes in ice tongue configuration." by Miles et al.**

September 22, 2020

**1 General comments**

In this manuscript, Miles et al reconstructed the migration of ice front and the evolution of the velocity field of Denman Glacier, Aurora subglacial basin, East Antarctica based on satellite images from 1962 to 2018. The ice sheet model Ua is then implemented to study the potential drivers of the widespread acceleration of Denman glacier between 1972 and 2009.

The manuscript is well written and easy to follow. Reconstruction of historical evolution of ice flow as well as the calving events is valuable for modellers to verify and improve the physical processes and parameterizatons in the ice sheet models. While the observation work is fascinating, I find the numerical modeling experiments not enough to

support the conclusions. The authors conclude that grounding line retreat, ice tongue thinning and unpinning from the pinning point are necessary to rebuild the acceleration over the glacier, and therefore emphasize the impact of ocean warming and calving events to the dynamics of Denman glacier. The three experiments are thinning the ice shelf, adjusting the bedrock elevation near the grounding line, and altering the bedrock elevation of the pinning point. Changing the bedrock elevation to mimic the same unpinning seen by the ice fractures is very tricky, since it affects the entire ice geometry for the wrong reasons. I think the ice sheet model Ua is capable to simulate what you want to investigate, but I suggest adaptations of the simulations (see specific comments). Therefore, I suggest a major change of the manuscript for the simulation sections.

**2   Specific comments**

- Line 15: It's mentioned several times in the manuscript the potential instability due to the retrograde slope. I think it could help the readers to understand the configuation better if the authors show in one of the figures (e.g. Fig. 1) a transection along the flow line to show the geometry.

- Line 17: ice tongue structure → ice-tongue structure. Please check the use of the hyphens over the manuscript.

- Line 21: In this manuscript, 'grounding line retreat' is taken as a driver of the recent change of Denman glacier. This is a confusing expression, grounding line retreat is a change (due to mass loss at the grounding line) rather than a driver of change. For example, continuous grounding line retreat on the retrograde slope is a result of ice shelf thinning and increasing ice flux at the grounding line, not the other way around.

- Line 30: Is 'Wilkes Land' here and also in Line 49 the same region as 'Aurusa Subglacial Basin'? If so, use one of them to avoid confusion.

- Line 68: mention also the conclusions section.

- Line 127-129: The authors predicted that calving event is unlikely due to the absence of any significant rifting or structural damage. The context of calving is missing in the introduction section, such as what could be the earlier indicators of calving events and how do we predict calving.

- Line 139-140: Could you make the scales of subplots of Figure 2 consistent to clearly show the information?

- Line 147-148: Are the rifts a indicator of calving events? Can you discuss more about the formation and development of the rifts? From Figure 2 it's hard for me to see.

- Line 151: 'Fid. 2d' → 'Fig. 2d'

- Line 159-161: In year 1984 there is a calving event. Do you think that could be one of the reasons that the speed-up is much higher between 1972-74 and 1989 while between 1989 and 2016-17 is slower?

- Line 180-182: Could you have another layer of ice flow magnitude and directions (arrows like Figure 5) on top to show the divergence of the ice flow?

- Line 202-204: 'Ice rheology is assumed to...'→ 'The relationship between creep and stress is assumed to...'

- Section 4.1: Modelling work is done to understand the acceleration/slowing down of the observed ice flow. Therefore, I think it's essential to at least show the momentum equations implemented by the ice sheet model, where the readers

could clearly see how ice geometry, basal sliding and ice rheology influence the velocity field.

- Section 4.2: The experiments are diagnostic based on the ice geometry from 2009 and the reconstructed ice geometry from 1972, is that right? This should be clarified.

- Experiment (i): The authors modified the ice-shelf thickness with an annual rate. How about the grounded ice upstream from the ice shelf? Are they kept the same between the two simulation years? Will it cause a dramatic thickness change near the grounding line in 1972? Or is there an interpolation done? Please describe your method. Could you show the geometry difference between the two simulations somewhere in the figures?

- Experiment (ii): I think it's not appropriate to call this experiment 'grounding line retreat', because grounding line retreat is impossible without ice geometry change. This experiment adjust the bedrock to have grounding line at a different location. How much uplifting is needed? Normally the bedrock won't have significant change in short term. The difference of velocity comes from additional basal friction in the uplifted region. This experiment actually shows the sensitivity of velocity field to the basal sliding near the grounding line.

- Experiment (iii): It's mentioned in the abstract and the discussion section that the unpinning of ice from Chugunov Island is due to the calving event. From Fig. 4d, e, and also mentioned in the results section, the ice around Chugunov Island might be heavily damaged, leading to unpinning/debuttressing. The calving effect and the damage effect could be simulated by changing the ice front or modify the rate factor 'A'. Why did the authors decide to evaluate the unpinning effect by changing bedrock? Furthermore, how much do you need to change to have the proper unpinning effect?

- Line 267: E1 is Fig. 5b, not Fig. 5a.

- Line 281: 'experiment 5' → 'experiment E5' or 'E5' same for the other experiments there after.

- Line 286-288: Can you explain why the acceleration on the ice shelf is much higher than the grounding line (Fig. 5a) but all simulations show the opposite pattern?

- Line 361: 'E3; Fig. 5e' → 'E3; Fig. 5d'?

- Line 362: 'E4; Fig.5d' → 'Fig. 5a'?

- Line 367: 'E4; Fig. 5d' → 'E4; Fig. 5e'?

- Line 379-381: Could the authors add the simulation of calving event by simply change the ice front position and evaluate its influence on ice velocity?

- Line 433: 'hydrofracturing' → 'hydrofracturing.'

- Line 447: Morlighem et al., 2019 is not in the reference list.

- Figure 1: Could the authors add the transection of the geometry along a flow line to clearly show the retrograde bed? Maybe show the perturbation of bedrock in experiment (ii) in the same plot. Point out the position of Chugunov Island.

- Figure 2: The subfigures are oriented in different ways, and with different scales, making it hard to compare the size of ice bergs, the development of rifts and so on. Could you have a zoom out subfigure like Figure 1b and put the boxes on top to show the zoom in area of the subfigures?

- Figure 4: Could you add a layer of velocity (magnitude and direction) on top of the satellite images to show the change of ice flow? That will make the figure more self-explanatory.

- Figure 5: I think there should be a grounding line contour at the pinning point Chugunov Island.

- Figure S2: Could you have a subfigure of simulated velocity? And also please show the location of the grounding line.

---

## Author Comment (AC1) · 21 Oct 2020

**Reviewer 1: Chad Greene**

General comments In this paper, Miles et al. generate observations of velocity and calving front positions of Denman Glacier, then they apply various perturbations to the geometry of a simple ice sheet model to determine what mechanisms might explain the observed behavior of the ice system. The study is elegantly designed and the manuscript is very well written.

The historical context provided by the ARGON and other early satellite photography is valuable, and I appreciate the work the authors have done to sift through the archives, in which they found a coherent and story to tell. I especially appreciate that the authors present background information in a way that sets the stage for understanding why this research was performed and what the results might mean for the future. The paper is packed with little insights such as the fascinating link between ice shelf thinning and flow direction, yet despite the density of information the text flows effortlessly. It was an enjoyable read, I learned a bit, and I recommend the paper for publication after the data and code are made publicly available.

*The authors thank Chad for taking the time to review our manuscript. We appreciate his positive comments and constructive suggestions throughout the review. We respond to each point below:*

**Data and code sharing:** This is important work, and in the future there will undoubtedly be more studies of the flow speed of Denman Glacier. Part of that work will involve reporting on changes that will have occurred since the publication of this study, and there's a good chance the authors of such a future study will want to begin by plotting velocity profiles from the 2020s on top of the results shown in Fig 3e. To allow others to build on this work, please include the coordinates and measured velocities shown in Fig 3e as supplemental material to the manuscript or upload to a data repository such as PANGAEA. Similarly, the authors have generated a wonderful calving front extent dataset shown in Fig 2b...Please share it so others may build on this work! Same goes for the Úa model code that was used to generate these results—I would love to see it after reading this paper.

*Thank-you for bringing this very important point to our attention. If the manuscript is accepted we will upload all data to the UK Polar Data Centre repository, this will include: All ice-front position shapefiles, historical velocity tifs, the coordinates and measured velocities in Figure 3e and the Ua code used to generate our simulations. The source code for Ua is already available at https://doi.org/10.5281/zenodo.3706624*

**L450-464:** At the end of a near-perfect manuscript in which each sentence brings new insights while setting the stage gracefully for the sentence that follows it, the final couple of paragraphs transition into a series of miscellaneous ideas that are related to, but not clearly relevant to the 2 main story of the manuscript. Each of these points could be expanded by a few sentences to help glue them to the findings of the study, but I don't think there's a need. Rather, most of the last two paragraphs could be deleted without detriment to the manuscript. I recommend simply reminding readers of the key historical behavior and/or future potential of Denman, and placing your results firmly in that context.

*In the revised version we have deleted the final paragraph of the manuscript. Our conclusion now simply reminds the reader of our key results and briefly notes the potential future vulnerability of Denman Glacier.*

Technical corrections

L26-27: The general sentiment of the final sentence of the abstract is reasonably supported by the analysis presented in this manuscript, but the phrase "...over the coming century" constrains the prediction a bit too tightly, because timescales of ice response are not directly discussed in this paper.

*Amended to: 'that it could be poised to make a significant contribution to sea level in the near future'*

L63: "...a range of remote sensing observations..." Make this sentence more clear by stating explicitly that velocity and ice front position observations are analyzed.

*We have amended the text to explicitly state velocity and ice front.*

L94-95: Indicate how the 1 pixel and 0.5 pixel error estimates were obtained. L113-114: Again, indicate how the error estimates were obtained.

*At the request of reviewer 2 we have included a more in depth analysis of velocity uncertainties from the historical 1970s data in supplementary figure 2. This is done by comparing manually tracked rift displacement at various locations across the Denman system to computed displacement values from the Cossi-Corr algorithm. The median value of the difference between the manually tracked rift displacement and the Cossi-Corr values is ±29 m yr$^{-1}$ (~0.5 pixels), thus justifying our estimated error.*

L113-114: Again, indicate how the error estimates were obtained.

*The error associated with ice-front position mapping of large Antarctic outlet glaciers has been established in several previous studies and we now make this clear in the text:*

*'Several previous studies (e.g. Miles et al., 2013; 2016; Lovell et al., 2017) have demonstrated that the errors associated with the manual mapping of ice-fronts from satellites with a moderate spatial resolution (10-250 m) are typically 1.5 pixels, with co-registration error accounting for 1 pixel and mapping error accounting for 0.5 pixels.'*

155-156: I don't think "accelerations" should be plural here. I recommend replacing "...with accelerations of 19±5%..." with "...with an overall acceleration of 19±5%..." Unless I've misunderstood the meaning of the sentence, in which case, please clarify.

*Amended.*

L246-252: I had to make this table to keep the experiments straight in my head. It may save others the same trouble to have the experiments explicitly tabulated in the manuscript.

*This is an excellent suggestion and we have added a similar table (Table 1) in the revised manuscript.*

L259: I think "each simulation" should be "each of the seven simulations".

*Amended.*

L330-331: Comparing accelerations as scalar multiples of each other is confusing, because I don't have any intuition for what it means if one thing has three times the acceleration of another thing. Actually, the sentence says "the ice accelerated approximately three times 3 faster" and if a is three times more than b, then a=4b, which makes the sentence even more confusing. Reword.

*We have amended to text to only refer to accelerations in percent:*

*'Between 1972 to 1990, observations indicate that ice accelerated 26 ±5% on the ice tongue (Fig. 3b) and 11 ±5% at the grounding line (Fig. 3c) in comparison to more limited accelerations of 9 ±1% and 3 ±2%, respectively, between 1990-2017'*

L424: Explicitly state what "this event" is. i.e., "Thus, the next major calving event..." And if the implications are important, don't just imply them—directly state what is implied are. i.e., "...could dictate the flow speed and direction of the..."

*We have amended the text to following:*

*'Thus, this calving event may have important implications for the evolution of the Denman/Shackleton system for multiple decades because it could influence both ice flow speed and direction.'*

Figure 1: This figure shows bedrock topography, ice velocity, and the spatial distribution of pinning points in the Shackleton Ice Shelf. These are all valuable as context for the study, and I appreciate that the figure legend clearly states the important things that viewers should take note of. My only complaint is that each variable is plotted in a separate panel, so understanding relationships between velocity, bed topography, and pinning points requires pinballing between all three panels as a way to mentally try to bring the variables all into one figure. Reconstituting the three variables is made more difficult by the fact that each panel shows different spatial extents, and at different scales. I recommend experimenting with transparency, vectors, or contours to show all three variables on one plot. That would also allow more detail, as a single panel could be enlarged to fill the entire width of the page. For example, something like the following would be a way to show ice velocity in the context of surface features and the bed topography that ties Denman Glacier to the ASB:

*We have revised figure 1 so that all three variables (velocity, bed topography and pinning points) are displayed on the same figure. On the basis of some of the other reviewer comments we also include a bed profile subplot taken over the Denman grounding line.*

Figure 2: As a logical sequence, I'd also put panel b before panel a, because currently panel b shows the direct observations and then panel a shows a quantified version of the observations. I'm also having difficulty understanding where panels c-f are in relation to panel b. There are no recognizable reference points in any of the images, so it's difficult to place them in space. I would typically assume that the image orientation remains constant across all panels, but the spatial extent and even the spatial scale is different in each panel, so everything is in question.

It's also tempting to assume panels c-f depict a sequence of events, but they are presented out of chronological order, so there's an extra little bit of mental bookkeeping that viewers must do to reconstruct what has happened to this glacier tongue since 1962.

If it makes sense to do so, I'd like to see the spatial extents of panels c-f remain constant across all panels, so it will be easy to follow changes over time. I suspect the entire figure would be easier to digest if the panels were rearranged, and if the times of panels c-f were marked directly on the ice front position time series. Something along these lines feels much more intuitive to me:

Or perhaps the time series plot would fit best below the calving-front map, but however you do it, I think the sequence of the panels is important for understanding what story is being told by the figure, and drawing direct connections between all the panels (such as by labeling the times of panels c-f directly on the time series) will help viewers see how the information is all related.

Also, more can be done in the caption to help readers understand the connection between ice front position and ice morphology. This could be just a sentence or two, but just something to help viewers see why R1 through R7 are labeled.

*In the revised figure with have added an additional panel (a) which provides a wider picture and highlight the location of the subsequent sub plots. We have also added a consistent reference grid to*

*all panels to provide a reference point on both the size and location of each figure. We have also added the times of the panels (d-g) to the ice-front time series.*

*We have also added more detail to the figure caption and included more detail on the importance of R1-R7:*

*'Figure 2: a) Overview of the Denman ice tongue with the coloured boxes indicating the locations of c-g. b) Reconstructed calving cycle of Denman Glacier 1940-2018. c) Examples of ice-front mapping 1962-2018. Note the change in angle of the ice shelf between its present (light blue – dark blue lines) and previous (pink-red lines) calving cycle. d) ARGON image of a large tabular iceberg in 1962 which likely calved from Denman at some point in the 1940s. e) Landsat-1 image of the Denman ice tongue in 1972, note the pattern of rifting labelled R1-R7. f) Landsat-4 image of a large tabular iceberg which calved from Denman in 1984. Note the rifting pattern and the absence of R7, meaning R7 likely propagated during its calving event in 1984. g) Landsat-8 image of the Denman ice tongue. Note the absence of rifting.'*

Figure 3: State which grounding line dataset is being shown here. Partly to give credit to the data producers, but also because InSAR and break-in-slope grounding lines don't agree here, and knowing which GL is plotted would help readers visually identify where certain features are relative to a particular GL

*The grounding line product used in Figure 3 is from Depoorter et al. (2013). We use this product for display purposes on the figure because it shows a clear and continuous grounding line across the study area. We have added the citation to the figure caption.*

Figure 5: The mental ledger keeping required to interpret this figure is not terribly onerous, but it involves more steps than are necessary. For example, if I want to know what's being depicted in panel g, I must go to the legend in the bottom left, where I see g corresponds to E6, then I find E6 in panel i, and then I say, "okay, E6 falls closer to a dashed line than most of the other dots do." And then I get curious about the outlier dot corresponding to E4 on the x axis. "I wonder what that is," I think, and so I repeat the process backward, going to the legend in the lower left of the figure to find that E4 corresponds to panel e, so then I look at panel e and I see a mostly blank white panel. At no point in that process is there any indication of what any of 7 these letters and numbers mean, because the figure has been stripped of all links to physical processes.

I recommend eliminating the legend from the bottom left and simply labeling "E1: ice shelf thinning," "E2: grounding line retreat," etc., directly on panels a-h, either as titles outside the plot or in the empty space in the bottom of each panel. That would also free up the text of the figure caption to focus on physical processes, rather than bookkeeping.

In the text caption, hammer home the main point by stating that E7 most closely matches observed velocities, suggesting that ice shelf thinning, grounding line retreat, and unpinning from Chugunov Island have all occurred since 1972.

Panel c shows the effects of grounding line retreat, but grounding line retreat itself is not shown. It's hard to gauge spatial scales here, but would a 10 km retreat be visible at this scale? If so, show both the 1972 and 2009 grounding lines.

Panel e shows the effects of unpinning from Chugunov Island. It would be helpful to label Chugunov Island directly on that panel.

I appreciate that panel i puts most meaningful region of velocities of each experiment in context with each other, while also showing the observed 1972 and 2009 velocities, but the panel comes up short in communicating the main point. It's relatively innocuous, so keep the panel if it feels important, but know that it adds a layer of complication to interpreting the figure as a whole. If you'd like to keep it, I recommend including Box D velocities from panel a as a data point. That would make it more clear how E4 got so out of line relative to the others. If you decide to eliminate panel i, the ice speed values could simply be printed next to Box D in their respective panels and/or included as a column in the table I recommended above.

I find myself leaning in close and squinting to see the details around the grounding line. Then I zoom the pdf to 300% and realize the problem isn't my eyesight, but the coarse resolution of the graphics. I recommend recreating the figure at higher resolution (If it's Matlab, try export_fig myfigure.png -r600 for 600 dpi) and enlarging the figure to fill the full width of the page so readers can see the beautiful details that are surely present in this data.

*We have added a text description of the experiment number and the perturbations forced in each panel of the figure to prevent the reader having to constantly flick between the caption and the figure. We have also labelled Chugunov Island on the appropriate figures. The scale is too coarse to the differing grounding line positions to be visible, but we do note that a close up version of the grounding line positions used in the simulations are in Fig S3. We have decided to keep panel I, which shows the velocities at box D to be consistent with the time series of speed change in Figure 3. We have also increased the resolution of the figure so more detail can be observed when zooming in and amended the figure caption.*

---

## Author Comment (AC2) · 21 Oct 2020

**Reviewer 2**

In this manuscript by Miles et al., the authors explore the connection between surface ice velocity acceleration and calving events for Denman Glacier, East Antarctica. Particularly, to explore this connection, the authors apply several tools ranging from remote sensing observations to ice sheet modelling to grasp which mechanisms might explain the acceleration and ground line retreat of Denman Glacier.

The manuscript is very well written and flows quite smoothly in the description of the methods and the used ice sheet models. As a remote sensing expert, I really liked the contribution brought by the historical remote sensing data, which are generally very difficult to find. The main hassle for me was going through all the figure. The absence of system coordinates makes very difficult to go from one figure to the other and make connections and comparisons between the figure. Therefore, I recommend the paper to be published after major revisions. In the following, some additional comments.

*We thank the reviewer for both the positive comments detailed above and for the constructive suggestions suggested below. We respond to each point detailed below.*

Line 11, identifying Denman glacier as the largest is really vague. For a reader, it would be nice to specify that e.g. this glacier is the largest contributor to sea level rise in East Antarctica (after Totten Glacier)

*We have amended the sentence to state that Denman is the largest contributor to sea level rise in East Antarctica after Totten Glacier.*

2. One of the major statements of the paper is that to explain the acceleration pattern of Denman Glacier it is required to have a combination of grounding line retreat, changes in ice shelf thickness and unpinning of ice from Chugonov Island (lines 331-334). I am wondering if the unpinning of Chugunov Island comes from observations. Did the authors observe the unpinning in their data?

*We do observe the unpinning of Denman's ice tongue from Chugunov Island. This is shown in Figure 4d & e and is described in the results section 3.3: 'Lateral migration of Denman's ice tongue':*

*'In 1974, the ice tongue was intensely shearing against Chugunov Island, as indicated by the heavily damaged shear margins (Fig. 4d). However, by 2002 the ice tongue made substantially less contact with Chugunov Island because this section of the ice tongue migrated westwards (Fig. 4d, e)'*

3. Line 94-95 and line 109-112. Have the authors quantitatively determined the uncertainty of surface ice velocity obtained from historical data? Instead of providing a ball park number, it would be nice to have a section in the supplementary describing how they quantified this uncertainty.

*We now include an additional figure in the supplement (Fig. S2) detailing the quantification of the velocity error for the historical imagery. This is done by comparing manually tracked rift displacement at various locations across the Denman system to computed displacement values from the Cossi-Corr algorithm. The median value of the difference between the manually tracked rift displacement and the Cossi-Corr values is ±29 m yr$^{-1}$ (~0.5 pixels), thus justifying our estimated error.*

4. To my understanding, to perform their analysis the authors have used the Measure grounding line product for Denman. Do you know to which sensor and which year this grounding line belongs to? Also, have the authors tried to include more recent grounding line products?

*We use the grounding line position from the BedMachine (v1) dataset, which in turn matches the grounding line position of the MEaSUREs dataset. The grounding line in the MEaSUREs dataset was derived from the ERS sensor in 1996. More recent grounding line products (~2017) over Denman have been made as of April 2020 (Brancato et al., 2020). We have not tried to include this more recent grounding line data into our models in order to retain consistency with the ice thickness calculated in the BedMachine product.*

5. I find really difficult to go through all the model experiments that the authors have performed. Is there a way to visually summarize them in the manuscript, e.g. using a table?

*We appreciate the multitude of modelling experiments is somewhat challenging to summarize in text, so we now include a table which visually summarizes the seven experiments (Table 1).*

6. To explain the acceleration of Denman glacier, in Section 5.2 the authors discuss whether this is driven by warm water intrusion (e.g. ice-ocean interactions) or calving events. I find this section a bit confusing especially because it introduces a lot of elements which have not previously discussed in the manuscript. Do the authors simulate with their ice model the effect of the ice-ocean interaction? Also, does the model take into account the effect of basal melt of Denman ice shelf?

*In our numerical modelling experiments we perturb both grounding line position and ice shelf thickness. These perturbations are extrapolated from modern observations of change (Ice shelf thickness; Paolo et al.; Grounding line migration; Brancato et al., 2020). Because both grounding line position and ice shelf thickness are sensitive to ice-ocean interaction, our model does indirectly simulate possible long-term effects of ice-ocean interaction.*

7. As a general comment, ALL figures need to have coordinates grid. For me it was extremely difficult to move from one

*We have added coordinate grids to all figures.*

8. I find the colorbar in figure 1-a to be confusing. Maybe all areas with an elevation over 0 should be marked with the same color, e.g. white.

9. Without coordinates is a bit difficult to link Fig.1b and Fig.1c. I would recommend to highlight the area in Fig.1c using a box in Fig.1b

*Point 8 &9: In response to the comments from all three reviewers we have made a new Figure 1. This revised figure 1 has reference coordinates and only highlights regions with a bed elevation below zero.*

10. Date of used Landsat images and acknowledgement for the used Landsat product should be included in the figure caption. Please, see USGS website for getting the proper data citation.

*We have added the dates and acknowledgement for the used Landsat products in the figure captions.*

11. For Fig.2, each panel focus on a different area of the ice shelf. I would recommend to have a bigger picture of the ice shelf on the side and try to help the reader understand on which areas you are focusing/zooming on. Please, include latitude and longitude (or polar sterographic) coordinates for each figure.

*We have amended figure 2 to include a reference image of the wider ice shelf (a) with boxes representing the location of each sub figure. We have also included polar stereographic coordinates on each image for reference.*

---

## Author Comment (AC3) · 21 Oct 2020

**Reviewer 3**

In this manuscript, Miles et al reconstructed the migration of ice front and the evolution of the velocity field of Denman Glacier, Aurora subglacial basin, East Antarctica based on satellite images from 1962 to 2018. The ice sheet model Ua is then implemented to study the potential drivers of the widespread acceleration of Denman glacier between 1972 and 2009.

The manuscript is well written and easy to follow. Reconstruction of historical evolution of ice flow as well as the calving events is valuable for modellers to verify and improve the physical processes and parameterizatons in the ice sheet models. While the observation work is fascinating, I find the numerical modeling experiments not enough to support the conclusions. The authors conclude that grounding line retreat, ice tongue thinning and unpinning from the pinning point are necessary to rebuild the acceleration over the glacier, and therefore emphasize the impact of ocean warming and calving events to the dynamics of Denman glacier. The three experiments are thinning the ice shelf, adjusting the bedrock elevation near the grounding line, and altering the bedrock elevation of the pinning point. Changing the bedrock elevation to mimic the same unpinning seen by the ice fractures is very tricky, since it affects the entire ice geometry for the wrong reasons. I think the ice sheet model Ua is capable to simulate what you want to investigate, but I suggest adaptations of the simulations (see specific comments). Therefore, I suggest a major change of the manuscript for the simulation sections.

*We thank the reviewer for taking the time to comment on our manuscript and for the positive comments detailed above. We address the reviewers concerns on our simulations below:*

Specific comments

Line 15: It's mentioned several times in the manuscript the potential instability due to the retrograde slope. I think it could help the readers to understand the configuation better if the authors show in one of the figures (e.g. Fig. 1) a transection along the flow line to show the geometry.

*We now include a transect along the flow line in the subpanel in Figure 1 to show the retrograde slope.*

Line 30: Is 'Wilkes Land' here and also in Line 49 the same region as 'Aurusa Subglacial Basin'? If so, use one of them to avoid confusion.

*Wilkes Land is the geographical region, whereas the Aurora Subglacial Basin is the subglacial basin within Wilkes Land. We have amended the text to avoid confusion:*

*'This has raised concerns about the future stability of some major outlet glaciers along the Wilkes Land coastline that drain the Aurora Subglacial Basin (ASB)'*

Line 68: mention also the conclusions section.

*We have added reference to the conclusion in the text.*

Line 127-129: The authors predicted that calving event is unlikely due to the absence of any significant rifting or structural damage. The context of calving is missing in the introduction section, such as what could be the earlier indicators of calving events and how do we predict calving.

*We appreciate the reviewers point here. However, we note that the concept of the prediction of Denman's next calving event is only a very small part of the manuscript. We feel that adding*

*background information on the early indicators of calving events in the introduction would distract from the manuscripts main aim – to explore the drivers of Denman's acceleration since 1972.*

Line 139-140: Could you make the scales of subplots of Figure 2 consistent to clearly show the information?

*This is a good suggestion. We have revised Figure 2 to make the scales of the subplots in figure 2 consistent.*

Line 147-148: Are the rifts a indicator of calving events? Can you discuss more about the formation and development of the rifts? From Figure 2 it's hard for me to see.

*We hypothesize that the rifting observed on the Denman ice tongue in the 1970s was important in Denman's calving in 1984. The is because an analysis of the rift patterns on the ice tongue in 1974 and on the subsequent grounded iceberg in 1990 show that the iceberg calved from Rift 7 (See Fig. 2e,f). However, further discussion on the formation and development of the rifts is tricky. This is because we only have satellite imagery available in 1972 and 1974, the next full available image over the Denman ice tongue is not until 1989. Therefore, commenting further on the formation and development of these rifts is very difficult without a greater density of observations.*

*We have clarified this in the text:*

*'The rifts periodically form ~10 km inland of Chugunov Island (Fig. 2e), on the western section of the ice tongue, before being advected down-flow. But a more detailed analysis of how the rifts form is not possible because of the limited available satellite imagery in the 1970s and 80s.'*

Line 151: 'Fid. 2d' → 'Fig. 2d'

*Amended.*

Line 159-161: In year 1984 there is a calving event. Do you think that could be one of the reasons that the speed-up is much higher between 1972-74 and 1989 while between 1989 and 2016-17 is slower?

*This is an interesting point. If the ice calved in 1984 provided buttressing, a speed up after the calving event would be expected. However, we note that changes in ice shelf extent have not been a dominant driver in the longer-term speed up of Denman. This is because the ice front was further advanced in 2018 than it was in 1972, but ice at Denman's grounding line is flowing 17% faster in 2018 than it did in 1972. We think the main importance of the calving event in the longer-term evolution of Denman, is that it enabled ice to re-advance at a different angle and make less contact the pinning point.*

*In order to test the direct importance of the calving event on ice velocity we would ideally need satellite image pairs either side of the event, but unfortunately such imagery is not available. In the absence of satellite imagery, we could simulate the calving event by altering the ice-front position in Úa. We agree that this would be an interesting experiment, but we note that we already show seven numerical modelling experiment and that adding a further perturbation may add further complication to an already busy manuscript. Indeed, we note our justification for choosing to compare ice geometries in 1972 and 2009 is that observations show that ice front position was similar in each time period.*

*We have added further clarification in the text at line 225:*

*'We chose 2009 for this baseline setup, because the calving front is in approximately the same position as in 1972 when our glacier observations start, thus ruling out any acceleration is response to a change in ice-front extent'*

Line 180-182: Could you have another layer of ice flow magnitude and directions (arrows like Figure 5) on top to show the divergence of the ice flow?

*We have added the MEAsUREs velocity magnitude and direction to the overview figure (4a). We do not add the change in direction in flow due to the patchy nature of the data in the 1970s.*

Line 202-204: 'Ice rheology is assumed to...'→ 'The relationship between creep and stress is assumed to...'

*Amended.*

Section 4.1: Modelling work is done to understand the acceleration/slowing down of the observed ice flow. Therefore, I think it's essential to at least show the momentum equations implemented by the ice sheet model, where the readers C4 TCD Interactive comment Printer-friendly version Discussion paper could clearly see how ice geometry, basal sliding and ice rheology influence the velocity field.

*We have added the following to line 192:*

*Úa is used to solve the equations of the shallow-ice stream or `shelfy-stream' 193 approximation, (SSA , Cuffey & Paterson, 2010). This can be expressed for one horizontal dimension as :*

$$2\partial_x \left( A^{-1/n} h \left( \partial_x u \right)^{1/n} \right) - \cdot \mathbf{G} \qquad C^{-1/m} u^{1/m} = \rho g h \partial_x s + \frac{1}{2} g h^2 \partial_x \rho$$

$$\rho$$

*Where A is the rate factor with its corresponding stress factor n, h is the vertical ice thickness, G is a grounding/flotation mask (1 for grounded ice, 0 for floating ice), C is the basal slipperiness with its corresponding stress exponent, m, ρ is the density of ice and g is the acceleration due to gravity.*

Section 4.2: The experiments are diagnostic based on the ice geometry from 2009 and the reconstructed ice geometry from 1972, is that right? This should be clarified.

*For clarification, Line 223 has been changed to:*

*"To ascertain the most likely causes of the observed acceleration for Denman ice shelf we start from a baseline set-up representing the ice shelf in 2009 where both ice geometry and velocity are well known and compare to diagnostic simulations of reconstructed 1972 ice geometry"*

Experiment (i): The authors modified the ice-shelf thickness with an annual rate. How about the grounded ice upstream from the ice shelf? Are they kept the same between the two simulation years? Will it cause a dramatic thickness change near the grounding line in 1972? Or is there an interpolation done? Please describe your method. Could you show the geometry difference between the two simulations somewhere in the figures?

*The thinning is applied only to fully floating nodes with grounded ice kept constant between simulations in a similar methodology used in Gudmundsson et al. 2019. As only fully floating are modified in this way, the thickness at the grounding line itself remains unmodified between the two simulations. Supplementary Figure 4 shows the thickness change applied, with a ~10 m thickening in the vicinity of the grounding line. The paragraph beginning line 235 has been changed to the following to include these additional clarifications.*

*"To represent ice shelf thinning since 1972, we take the mean annual rate of ice-thickness change from an 1994–2012 ice-shelf thickness change dataset (Paolo et. al., 2015) and scale it up to represent the total thickness change over the 37 years between 1972 and 2009, assuming that the 1994–2012 mean annual rate remains constant during this period. This thickness change is then applied to the 2009 ice geometry, modifying it to better represent the estimated 1972 ice thickness distribution of the Shackleton Ice Shelf, Denman ice tongue and Scott Glacier. Similar to the methodology of Gudmundsson et al. 2019, we only apply this thickness change to fully floating nodes, with no change of ice thickness for grounded ice and ice directly over the grounding line. The total thickness change applied is shown in Supplementary Figure 4. We refer to this perturbation as 'ice shelf thinning' because the majority of the floating portions of Denman's ice tongue and Shackleton Ice Shelf have thinned since 1994, although some sections of Scott Glacier have actually thickened near its calving front (Fig. S4)."*

Experiment (ii): I think it's not appropriate to call this experiment 'grounding line retreat', because grounding line retreat is impossible without ice geometry change. This experiment adjust the bedrock to have grounding line at a different location. How much uplifting is needed? Normally the bedrock won't have significant change in short term. The difference of velocity comes from additional basal friction in the uplifted region. This experiment actually shows the sensitivity of velocity field to the basal sliding near the grounding line.

*Our methodology is not designed to represent any real earth processes such as isostatic rebound but is instead intended to show the instantaneous effect of a grounding line perturbation on ice velocity with the minimum possible bias to the existing ice velocity field. Directly modifying the ice geometry at the grounding line will have a noticeable effect on the regional ice velocity field due to conservation of flux in addition to any changes arising from the shift in grounding line position, and so we instead raise the bedrock to force the models grounding line to be at a given location. The paragraph starting line 247 has been modified to clarify this:*

*"In the Úa ice model, the grounding line position is not explicitly defined by the user but is instead a direct result of ice thickness, bedrock depth and the relative densities of ice and sea water. As such, the two ways to perturb a given grounding line are to either modify the ice thickness or the bedrock depth. Modifying the bedrock depth is the less disruptive approach because the resulting effect upon velocity is not biased by an imposed change in ice thickness at the grounding line effecting the regional ice velocity field due to flux conservation, in addition to that caused by shifting the grounding line. Note that raising the bedrock to meet the underside of the ice shelf in this way is not a representation of any real earth processes, it is merely forcing the model to have the grounding line in a particular location, that than enables a diagnostic simulation. To represent grounding line retreat since 1972 we advanced Denman's grounding line from its position in the 2009 baseline set-up by 10 km to a possible 1972 position. This is achieved via raising the bedrock approximately ~20-30 m in the area shown in Fig. S4. We justify a 10 km retreat since 1972 based on the rate of grounding-line retreat observed between 1996 and 2017 (~5km; Brancato et al., 2020). For the newly grounded area, values of the bed slipperiness, C, are not generated in our model inversion, we therefore prescribe nearest-neighbour values to those at the grounding line in the model inversion."*

Experiment (iii): It's mentioned in the abstract and the discussion section that the unpinning of ice from Chugunov Island is due to the calving event. From Fig. 4d, e, and also mentioned in the results

section, the ice around Chugunov Island might be heavily damaged, leading to unpinning/debuttressing. The calving effect and the damage effect could be simulated by changing the ice front or modify the rate factor 'A'. Why did the authors decide to evaluate the unpinning effect by changing bedrock? Furthermore, how much do you need to change to have the proper unpinning effect?

*We agree with the reviewers point that the unpinning from Chugunov Island could be investigated by modifying the rate factor 'A', in addition to the regrounding experiment shown. Due to the limited information about past conditions of ice geometry and properties available to us, any attempt to simulate past conditions will have to some extent rely upon assumptions. We would argue that the assumptions made for the regrounding of the ice at Chugunov Island are more justifiable than those that would be required for an investigation into the rate factor, 'A'. Performing new model inversions using 1972 velocities and ice geometry would be the ideal way to investigate this. However, as the 1972 velocities are relatively patchy and the 1972 ice geometry itself an unknown under investigation it would be impossible to separate out the effect of the damaged ice on both velocity and rate factor from that arising from ice geometry change. The assumptions needed to investigate the effect of regrounding the ice at Chugunov island are easier to justify. Raising the bedrock by ~30m is enough to ground ice along the edge of the model domain and we have assumed basal slipperiness, C, is the same as that located near the grounding line. The paragraph beginning line 263 has been modified to further articulate our reasoning:*

*"To represent the pinning of Denman's ice tongue against Chugunov Island in the 1972 observations (e.g. Fig. 4d, e), we artificially raise a small area of bedrock on the western edge of Chugunov Island (Fig. S3). Bed slipperiness was set to a value comparable to that immediately upstream of the grounding line. Note that, although past observations suggest that the ice in front of Chugunov Island has been damaged, possibly having an effect on its rate factor, A, we have decided to limit our investigation to the effect of pinning the ice on Chugunov Island without changing rate factor. To properly investigate the possible change in past rate factor we would need less spatially patchy 1972 velocities as well as an accurate understanding of past ice geometry (itself an unknown under investigation) to perform a model inversion for 1972 conditions."*

Line 267: E1 is Fig. 5b, not Fig. 5a.

*Amended*

Line 281: 'experiment 5' → 'experiment E5' or 'E5' same for the other experiments there after.

*Amended*

Line 286-288: Can you explain why the acceleration on the ice shelf is much higher than the grounding line (Fig. 5a) but all simulations show the opposite pattern?

*For most of these simulations a direct comparison can be potentially misleading as we are applying the perturbations in isolation of one another to investigate the general pattern of change in velocity association with each individual perturbation. For example, the isolated un-pinning experiment (Fig. 5e) clearly shows that unpinning from Chugunov Island has a negligible effect on grounding line velocity. The difference from observations for most of these simulations is probably due to the simulations not only omitting a perturbation in ice geometry but also the interactions between different types of perturbation. For the simulation which includes all three perturbations (Fig. 5h) the difference from observations is noticeably less than in the isolated perturbations, and remaining differences can most likely be attributed to the uncertainties in the perturbations applied (eg.,*

*assuming the thickness change over the 37 years is the same as the annual scaled mean change between 1994—2012).*

Line 361: 'E3; Fig. 5e' → 'E3; Fig. 5d'?

*Amended*

Line 362: 'E4; Fig.5d' → 'Fig. 5a'?

*Amended*

Line 367: 'E4; Fig. 5d' → 'E4; Fig. 5e'?

*Amended*

Line 379-381: Could the authors add the simulation of calving event by simply change the ice front position and evaluate its influence on ice velocity?

*Our observations show that changes in ice-front extent was unlikely to have driven the long-term acceleration of Denman. This is because that Denman's ice-front is currently further advanced than it was in 1972, but we still observe a 17% acceleration in flow over the grounding line. Therefore, we do not simulate the impact of the calving event by changing the ice-front position because our observations demonstrate that this is not an important contributor to the long-term acceleration of Denman*

*In our description of the perturbation experiments we clarify our rationale for not simulating the change in ice-front position (Line 225):*

*'We chose 2009 for this baseline setup, because the calving front is in approximately the same position as in 1972 when our glacier observations start, thus ruling out any acceleration is response to a change in ice-front extent'*

Line 433: 'hydrofracturing' → 'hydrofracturing.'

*Amended.*

Line 447: Morlighem et al., 2019 is not in the reference list.

*Amended. The citation in the main text should read 'Morlighem et al., 2020' – we have corrected throughout.*

Figure 1: Could the authors add the transection of the geometry along a flow line to clearly show the retrograde bed? Maybe show the perturbation of bedrock in experiment (ii) in the same plot. Point out the position of Chugunov Island.

*This is a good suggestion and we have added a transect of the bedrock elevation along the flow line in a subplot to show the retrograde slope.*

Figure 2: The subfigures are oriented in different ways, and with different scales, making it hard to compare the size of ice bergs, the development of rifts and so on. Could you have a zoom out subfigure like Figure 1b and put the boxes on top to show the zoom in area of the subfigures?

*We have revised Figure 2 to include a zoom out subfigure with boxes indicating the location of the other subpanels. We have also added reference coordinates to all panels and made the scales consistent to enable an easier comparison of the icebergs and rifts.*

Figure 4: Could you add a layer of velocity (magnitude and direction) on top of the satellite images to show the change of ice flow? That will make the figure more self-explanatory.

*We have added velocity magnitude and direction to the overview figure (4a). We do not add the change in direction in flow due to the patchy nature of the data in the 1970s.*

Figure 5: I think there should be a grounding line contour at the pinning point Chugunov Island.

*We have added a grounding line contour around Chugunov Island on the appropriate panels.*

Figure S2: Could you have a subfigure of simulated velocity? And also please show the location of the grounding line.

*Figure S2 has had the recommended changes made.*